# Operation regimes of spinal circuits controlling locomotion and the role of supraspinal drives and sensory feedback

**Ilya A Rybak[1]\*, Natalia A Shevtsova[1], Sergey N Markin[1], Boris I Prilutsky[2], Alain Frigon[3]\***

[1]Department of Neurobiology and Anatomy, College of Medicine, Drexel University, Philadelphia, United States; [2]School of Biological Sciences, Georgia Institute of Technology, Atlanta, United States; [3]Department of Pharmacology-Physiology, Faculty of Medicine and Health Sciences, Centre de Recherche du CHUS, Université de Sherbrooke, Sherbrooke, Canada

**\*For correspondence:**
iar22@drexel.edu (IAR);
Alain.Frigon@USherbrooke.ca
(AF)

**Competing interest:** The authors declare that no competing interests exist.

**Abstract** Locomotion in mammals is directly controlled by the spinal neuronal network, operating under the control of supraspinal signals and somatosensory feedback that interact with each other. However, the functional architecture of the spinal locomotor network, its operation regimes, and the role of supraspinal and sensory feedback in different locomotor behaviors, including at different speeds, remain unclear. We developed a computational model of spinal locomotor circuits receiving supraspinal drives and limb sensory feedback that could reproduce multiple experimental data obtained in intact and spinal-transected cats during tied-belt and split-belt treadmill locomotion. We provide evidence that the spinal locomotor network operates in different regimes depending on locomotor speed. In an intact system, at slow speeds (<0.4 m/s), the spinal network operates in a non-oscillating state-machine regime and requires sensory feedback or external inputs for phase transitions. Removing sensory feedback related to limb extension prevents locomotor oscillations at slow speeds. With increasing speed and supraspinal drives, the spinal network switches to a flexor-driven oscillatory regime and then to a classical half-center regime. Following spinal transection, the model predicts that the spinal network can only operate in the state-machine regime. Our results suggest that the spinal network operates in different regimes for slow exploratory and fast escape locomotor behaviors, making use of different control mechanisms.

## eLife assessment

This **fundamental** state-of-the-art modeling study explores neural mechanisms underlying walking control in cats, demonstrating the probability of three different states of operation of the spinal circuitry generating locomotion at different speeds. The authors' biophysical modeling sufficiently reproduces and provides explanations for experimental data on how the locomotor cycle and phase durations depend on treadmill walking speed and points to new principles of circuit functional architecture and operating regimes underlying how spinal circuits interact with supraspinal signals and limb sensory feedback signals to produce different locomotor behaviors at different speeds, which are major unresolved problems in the field. The modeling evidence is **compelling**, especially in advancing our understanding of locomotion control mechanisms and will interest neuroscientists studying the neural control of movement.

## Introduction

Locomotion is generated and controlled by three main neural components that interact dynamically (*Rossignol et al., 2006*; *Frigon et al., 2021*; *Frigon, 2017*; *Grillner, 1981*; *Kiehn, 2016*; *Orlovsky et al., 1999*). The spinal network, including circuits of the central pattern generator (CPG), generates the basic locomotor pattern, characterized by alternation of flexor and extensor activity in each limb and coordination of activities related to left and right limbs. Supraspinal structures initiate and terminate locomotion and control voluntary aspects of locomotion. Somatosensory feedback from primary afferents originating in muscle, joint, and skin mechanoreceptors provides information on the state of the musculoskeletal system and the external environment. Although we know that the spinal network generates the basic locomotor pattern, its functional architecture, operating regimes, and the way it interacts with supraspinal signals and somatosensory feedback to produce different locomotor behaviors, including at different speeds, remains poorly understood.

In terrestrial locomotion, the step cycle of each limb consists of two main phases, swing, and stance, that correspond primarily to the activity of limb flexors and extensors, respectively. The main role of corresponding spinal rhythm-generating circuits is to establish the frequency of oscillations, or cycle duration, and the durations of flexor and extensor phases (*McCrea and Rybak, 2008*). In mammals, including humans, increasing walking speed leads to a decrease in cycle duration mostly because of a decrease in stance/extensor phase duration while swing/flexor phase duration remains relatively unchanged (reviewed in *Gossard, 2011*; *Frigon, 2012*; *Halbertsma, 1983*). This is observed on a treadmill in intact animals and also following complete spinal thoracic transection (spinal animals). In intact animals, supraspinal signals interact with spinal networks and sensory feedback from the limbs, whereas in spinal animals, supraspinal signals are absent.

As an experimental basis of this study, we used data previously obtained in intact and spinal cats stepping on a treadmill with two independently controlled belts in tied-belt (equal speeds of left and right belts) and a split-belt (different speeds of left and right belts) conditions (*Frigon et al., 2015*; *Frigon et al., 2017*; *Latash et al., 2020*). Changes in cycle and phase durations during tied-belt and split-belt locomotion with increasing speed and left-right speed differences were similar in intact and spinal cats over a range of moderate speeds (0.4–1.0 m/s). This is somewhat surprising because intact and spinal cats rely on different control mechanisms. Intact cats walking freely on a treadmill engage vision for orientation in space and their supraspinal structures process visual information and send inputs to the spinal cord to control locomotion on a treadmill that maintains a fixed position of the animal relative to the external space. Spinal cats, whose position on the treadmill relative to the external space is fixed by an experimenter, can only use sensory feedback from the hindlimbs to adjust locomotion to the treadmill speed. An interesting additional observation here is that intact cats cannot consistently perform treadmill locomotion at very slow speeds (below 0.3–0.4 m/s), whereas spinal cats have no problem walking at such slow speeds (*Frigon et al., 2017*; *Dambreville et al., 2015*). This is evidently context-dependent and specific for treadmill locomotion as cats, humans, and other animals can voluntarily decide to perform consistent overground locomotion at slow speeds.

To investigate the organization and operation of spinal circuits and how different mechanisms interact to control locomotion, we developed a tractable computational model of these circuits operating under the control of both supraspinal drives and sensory feedback. Our model followed the commonly accepted concept that the spinal cord contains CPG circuits that can intrinsically generate locomotor-like oscillations without rhythmic external inputs (*Grillner, 1981*; *Kiehn, 2016*; *Orlovsky et al., 1999*; *McCrea and Rybak, 2008*; *Brown, 1911*; *Brown, 1914*; *Grillner and Zangger, 1975*; *Grillner and Zangger, 1979*; *Rossignol, 1996*; *Grillner and El Manira, 2020*). The concept of spinal mechanisms that intrinsically generate the basic locomotor pattern (CPG prototype) was initially proposed by *Brown, 1911*; *Brown, 1914* in opposition to the previously prevailing viewpoint of Maurice Philippson and Charles Sherrington (*Sherrington, 1910a*; *Sherrington, 1910b*; *Philippson, 1905*) that locomotion is generated through a chain of reflexes, i.e., critically depends on limb sensory feedback (reviewed in *Stuart and Hultborn, 1882*; *Clarac, 2008*).

The present model is based on our previous models (*Rybak et al., 2015*; *Shevtsova et al., 2015*; *Zhang et al., 2022*; *Shevtsova et al., 2022*; *Danner et al., 2016*; *Danner et al., 2017*; *Danner et al., 2019*) and contains two rhythm generators (RGs) that control the left and right hindlimbs and interact through a series of commissural pathways. The RGs receive supraspinal drive, and limb sensory feedback during the extension phase. Each RG consists of flexor and extensor half-centers operating as

conditional bursters (i.e. capable of intrinsically generating rhythmic bursting in certain conditions) and inhibiting each other. However, each RG is considered not as a simple neuronal oscillator, but as a neural structure with a dual function: as a state machine, defining operation in each state/locomotor phase independent of the phase-transition mechanisms, and as an actual oscillator, defining the mechanisms of state/phase transitions. A state machine, also called a finite-state machine, is a behavior model that can operate in only one of a finite number of states at any given time and may change its state (performing a *state transition*) in response to an external input (*Wang and Tepfenhart, 2019*; *Hopcroft et al., 2000*). We propose and show that, depending on conditions, each RG can operate in three different regimes: a non-oscillating *state-machine* regime and in a *flexor-driven* and a *classical half-center* oscillatory regimes. In the full model, the generated locomotor behavior depends on the excitatory supraspinal drives to the RGs and on limb sensory feedback. The gain of limb sensory feedback in the intact model is suppressed by supraspinal drive via presynaptic inhibition (*Rudomin and Schmidt, 1999*; *Eccles et al., 1961*; *Fink et al., 2014*) and is released from inhibition following spinal transection.

Our model reproduces and proposes explanations for experimental data, including the dependence of main locomotor characteristics on treadmill speed in intact and spinal cats. Particularly, based on our simulations, we suggest that locomotion in intact cats at low speeds and in spinal cats at any speed is mainly controlled by limb sensory feedback, consistent with Philippson's and Sherrington's view (*Sherrington, 1910a*; *Sherrington, 1910b*; *Philippson, 1905*), whereas locomotion in intact cats at higher/moderate speeds is primarily controlled by the intrinsic oscillatory activity within the spinal network, supporting Brown's concept (*Brown, 1911*; *Brown, 1914*).

## Results

### Cycle and phase durations during tied-belt and split-belt locomotion in intact and spinal cats

#### Tied-belt locomotion

Our previous studies in intact (*Figure 1A*) and spinal (*Figure 1B*) cats showed a decrease in cycle duration with increasing speed due to a shortening of the stance phase with a relatively constant swing phase duration during tied-belt locomotion (*Frigon et al., 2015*; *Frigon et al., 2017*; *Latash et al., 2020*). This agrees with other studies in mammals (*Gossard, 2011*; *Frigon, 2012*; *Halbertsma, 1983*; *Nilsson et al., 1985*). Note that most studies of treadmill locomotion in intact cats have been performed at treadmill speeds at or above 0.4 m/s (as in *Figure 1A*). When placed on a treadmill moving at a lower speed (from 0.1 m/s to 0.3 m/s), these cats demonstrate inconsistent stepping by making a few steps alternating with periods of stopping and/or sitting. Thus, when comparing intact and spinal cats (*Figure 1C*), we highlight the following observations: (1) Intact cats do not step consistently during quadrupedal tied-belt locomotion on a treadmill if the speed is at or below 0.3 m/s, whereas spinal cats have no problem performing hindlimb locomotion from 0.1 to 0.3 m/s. (2) Speed-dependent changes in swing and stance phase durations from 0.4 to 1.0 m/s are qualitatively similar in intact and spinal cats, but stance duration in intact cats is usually a little longer. (3) With increasing treadmill speed, the duty factor (ratio between stance and cycle durations) in intact and spinal cats decreases towards 0.5 (equal swing and stance proportion). In spinal cats, the duty factor reaches 0.5 at approximately 0.8–0.9 m/s.

#### Split-belt locomotion

We used data from our previous studies in intact (*Figure 2A*) and spinal (*Figure 2B*) cats during split-belt locomotion, where the left (slow) hindlimb stepped at 0.4 m/s and right (fast) hindlimb stepped from 0.5 to 1.0 m/s (*Frigon et al., 2015*; *Frigon et al., 2017*; *Latash et al., 2020*). When comparing intact and spinal cats, we highlight the following observations: (1) Increasing the speed of the fast belt leads to a decrease in cycle duration in intact and spinal cats, but the slow and fast hindlimbs maintain equal cycle duration; (2) In the slow hindlimb, increasing the speed of the fast belt leads to a small decrease in stance and swing durations in intact cats and in the swing duration of spinal cats; (3) In the fast hindlimb, increasing the speed of the fast belt produces a decrease in the stance duration and an increase in swing duration in both intact and spinal cats. The duty factor reaches 0.5 at 0.9 m/s and

0.8 m/s in intact and spinal cats, respectively, and goes below this value at 0.9–1.0 m/s in spinal cats, where swing duration occupies a greater proportion of the cycle.

## Conceptual framework and model description

### Basic model architecture

In this study, we fully accept the concept that the mammalian spinal cord contains neural circuits (i.e. CPG) that can (in certain conditions) intrinsically (i.e. without rhythmic or patterned external inputs) generate the complex coordinated pattern of locomotor activity (*Grillner, 1981*; *Orlovsky et al., 1999*; *McCrea and Rybak, 2008*; *Brown, 1911*; *Grillner and Zangger, 1975*; *Grillner and Zangger, 1979*; *Jankowska et al., 1967*; *Grillner, 2006*). Following our previous computational models (*Rybak et al., 2015*; *Shevtsova et al., 2015*; *Zhang et al., 2022*; *Danner et al., 2016*; *Danner et al., 2017*; *Danner et al., 2019*), we assume that the circuitry of the spinal locomotor CPG includes (a) rhythm-generating (RG) circuits that control states and rhythmic activity of each limb, and (b) rhythm-coordinating circuits that mediate neuronal interactions between the RG circuits and define phase relationships between their activities (coupling, synchronization, alternation). Also, following the classical view, we assume that each RG consists of two half-centers that mutually inhibit each other and define the two major

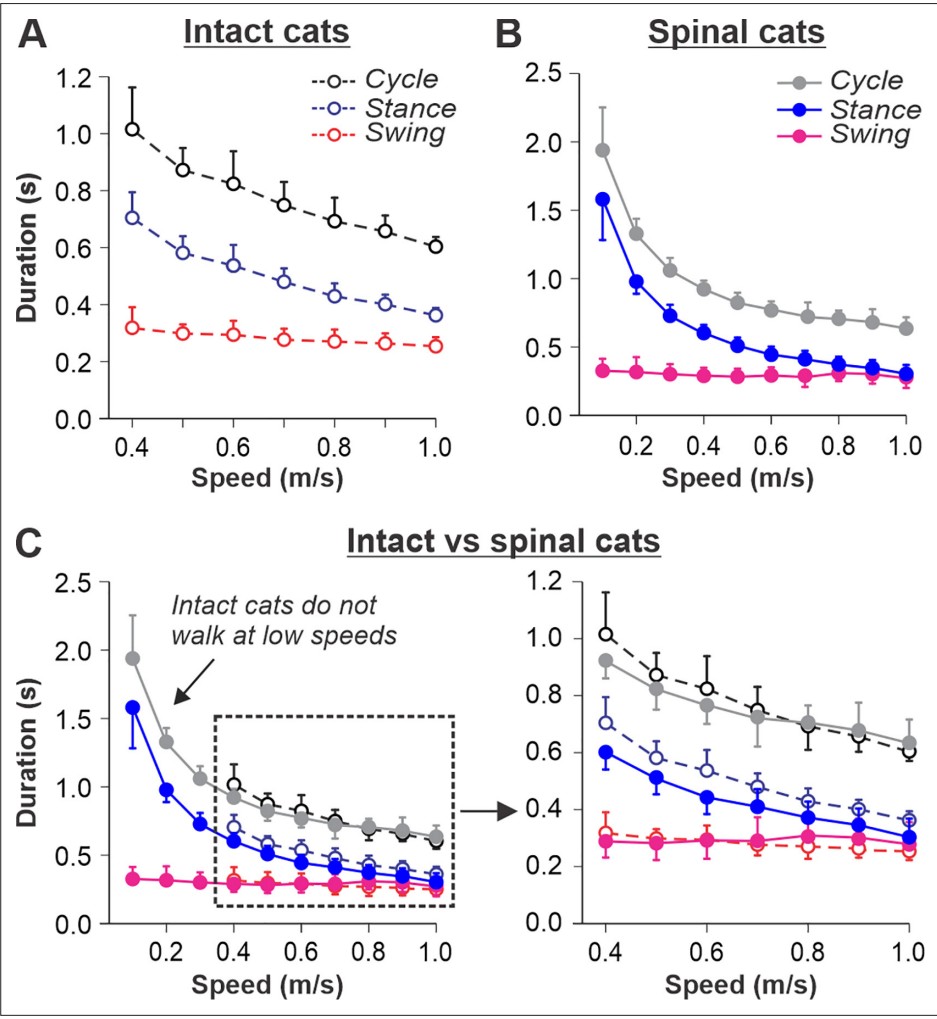

**Figure 1.** Locomotion of intact and spinal cats on a tied-belt treadmill. (**A, B**) Step cycle, stance, and swing phase durations for the right hindlimb during tied-belt treadmill locomotion of intact **A**, from *Frigon et al., 2015*; *Latash et al., 2020* and spinal **B**, from *Frigon, 2017*; *Latash et al., 2020* cats with an increasing treadmill speed. Data were obtained from 6 to 15 cycles in seven intact and six spinal cats (one cat was studied in both states). Each data point is the mean ± standard deviation. Modified from Fig. 3C, D of *Latash et al., 2020*, under the license CC-BY-4. (**C**) Superimposed curves from (**A**) and (**B**) to highlight differences.

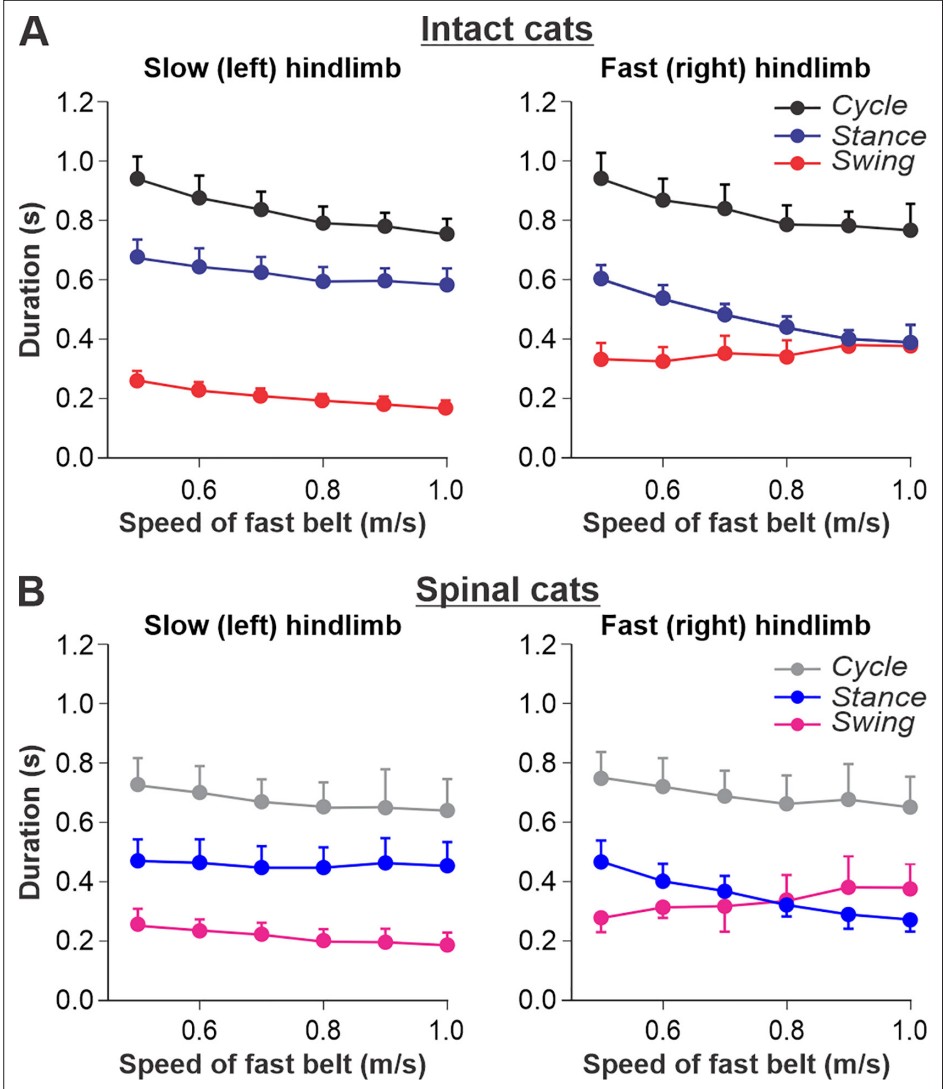

**Figure 2.** Locomotion of intact and spinal cats on a split-belt treadmill. (**A**) Step cycle, stance, and swing phase durations for the left (slow) and right (fast) hindlimbs during split-belt treadmill locomotion of intact cats (from *Frigon et al., 2015*; *Latash et al., 2020*). (**B**) Changes in the same characteristics for the left (slow) and right (fast) hindlimbs during split-belt treadmill locomotion in spinal cats (from *Frigon et al., 2017*; *Latash et al., 2020*). In both series of experiments the left (slow) hindlimb was stepping at 0.4 m/s while the right (fast) hindlimb stepped with speeds from 0.5 to 1.0 m/s with 0.1 m/s increments. Data were obtained from 6 to 15 cycles in seven intact and six spinal cats (one cat was studied in both states). Each data point is the mean ± standard deviation. Modified from Figure 6A and B of *Latash et al., 2020*, under the license CC-BY-4.

states/phases of the RG, the flexor and extensor phases, in which the corresponding sets of limb muscles are activated (*Orlovsky et al., 1999*; *McCrea and Rybak, 2008*; *Brown, 1911*; *Jankowska et al., 1967*).

## Modeling of a single RG and its operation regimes

The ability of an RG to generate rhythmic activity can be based on the intrinsic rhythmic bursting properties of one or both half-centers or can critically depend on the inhibitory interactions between half-centers, so that each half-center cannot intrinsically generate rhythmic bursting, as in the classical half-center concept (*Brown, 1911*; *Brown, 1914*). However, in the isolated mouse spinal cord using optogenetic stimulation, rhythmic activity can be induced independently in flexor- and extensor-related spinal circuits (*Hägglund et al., 2013*), suggesting that both flexor and extensor half-centers

can, in certain conditions, generate independent rhythmic bursting activity (i.e. operate as conditional bursters). Previous mathematical models of neurons defined and analyzed the conditions allowing these neurons to generate rhythmic bursting based on nonlinear voltage-dependent properties of different ionic channels (*Wang and Rinzel, 1995*; *Izhikevich, 2000*; *Izhikevich, 2006*; *Guckenheimer et al., 1997*; *Rinzel and Ermentrout, 1998*). Several models that described bursting activity in neurons of the medullary pre-Bötzinger complex for respiration or in the spinal cord for locomotion, suggested that this activity is based on a persistent (slowly-inactivating) sodium current, $I_{NaP}$ (*Rybak et al., 2015*; *Brocard et al., 2013*; *Rybak et al., 2004*; *Butera et al., 1999*; *Rybak et al., 2006a*; *Rybak et al., 2006b*; *Ausborn et al., 2018*). We implemented a similar $I_{NaP}$-dependent mechanism for conditional bursting in our RG half-center model (see Methods).

*Figure 3* illustrates the behavior of a neuron model, describing a single conditional burster (*Figure 3A*) with the output representing changes in spike frequency (see Methods). At the low excitatory drive, the neuronal output is equal to zero ('*silence*'), but when the drive exceeds some threshold, the model switches to a '*bursting*' regime, during which the frequency of bursts increases with increasing drive (*Figure 3B*). At a certain drive, the model switches to sustained '*tonic*' activity.

Let us now consider a simple half-center RG consisting of two conditional bursters, the flexor half-center (F), receiving an increasing Drive-F, and the extensor half-center (E), receiving a constant Drive-E that maintains it in a state of tonic activity if uncoupled. The two half-centers inhibit each other through inhibitory neurons, InF and InE (*Figure 3C*). At low Drive-F there are no oscillations, and the F half-center remains silent, although it could start oscillating if it was not strongly inhibited by the E half-center. The E half-center shows tonic activity because of the high value of Drive-E. We call this regime of RG operation a *state-machine* regime (*Figure 3D*, left part of the graph). In this regime, the RG maintains a state of extension, until an external signal, sufficiently strong, activates the F half-center or inhibits the E half-center (green arrows in *Figure 3C*) to release the F half-center from inhibition, allowing it to generate intrinsic bursts, switching the model to a flexion state. Increasing the excitatory Drive-F releases the F half-center from the E half-center's inhibition (acting via the InE neuron), switching the RG to a bursting regime. Note that in this regime, the E half-center also exhibits bursting activity in alternation with the F half-center due to rhythmic inhibition from the flexor-half center via the InF neuron, although the E half-center itself, if uncoupled, operates in the tonic mode. We call this regime a *flexor-driven* regime (*Figure 3D*, middle part of the graph). Similar to the intrinsic bursting regime in an isolated conditional burster (*Figure 3B*), with an increase in Drive-F, the bursting frequency of the RG is increasing (and the oscillation period is decreasing) mostly due to the shortening of the extensor bursts with much less reduction in flexor burst duration. Further increasing the excitatory Drive-F leads to a transition of the RG to a *classical half-center* regime (*Figure 3D*, right part of the graph), in which the half-centers cannot generate oscillations if uncoupled, and RG oscillations occur due to, and critically depend on, mutual inhibition between half-centers and an adaptive reduction of their responses. In this regime, with increasing Drive-F, the oscillation frequency (and period) remains almost unchanged, and flexor burst duration increases to compensate for decreasing extensor bursts. To summarize, increasing the excitatory drive to the flexor half-center in this simple RG model demonstrates a sequential transition from a *state-machine* regime to a *flexor-driven* regime, and then to a *classical half-center* regime. The sensitivity analysis of this model can be found in *Figure 3—figure supplement 1*.

## Model of the spinal locomotor circuitry

The schematic of our model is shown in *Figure 4*. The model incorporates neuronal circuits involved in the control of, and interactions between, two cat hindlimbs during locomotion on a treadmill. The spinal circuitry in the model includes two RGs (as described above) interacting via a series of commissural interneuronal (CIN) pathways mediated by different sets of genetically identified commissural (V0$_D$, V0$_V$, and V3) and ipsilaterally projecting (V2a) interneurons, as well as some hypothetical inhibitory interneurons (Ini). The organization of these intraspinal interactions was directly drawn from our earlier models (*Rybak et al., 2015*; *Shevtsova et al., 2015*; *Zhang et al., 2022*; *Shevtsova et al., 2022*; *Danner et al., 2016*; *Danner et al., 2017*; *Danner et al., 2019*) that were explicitly or implicitly based on the results of molecular/genetic studies of locomotion in mice or were proposed to explain and reproduce multiple aspects of the neural control of locomotion in these studies. Specifically, V0$_D$ and V2a-V0$_V$-Ini pathways secure left-right alternation of RG oscillations during walking and trotting at

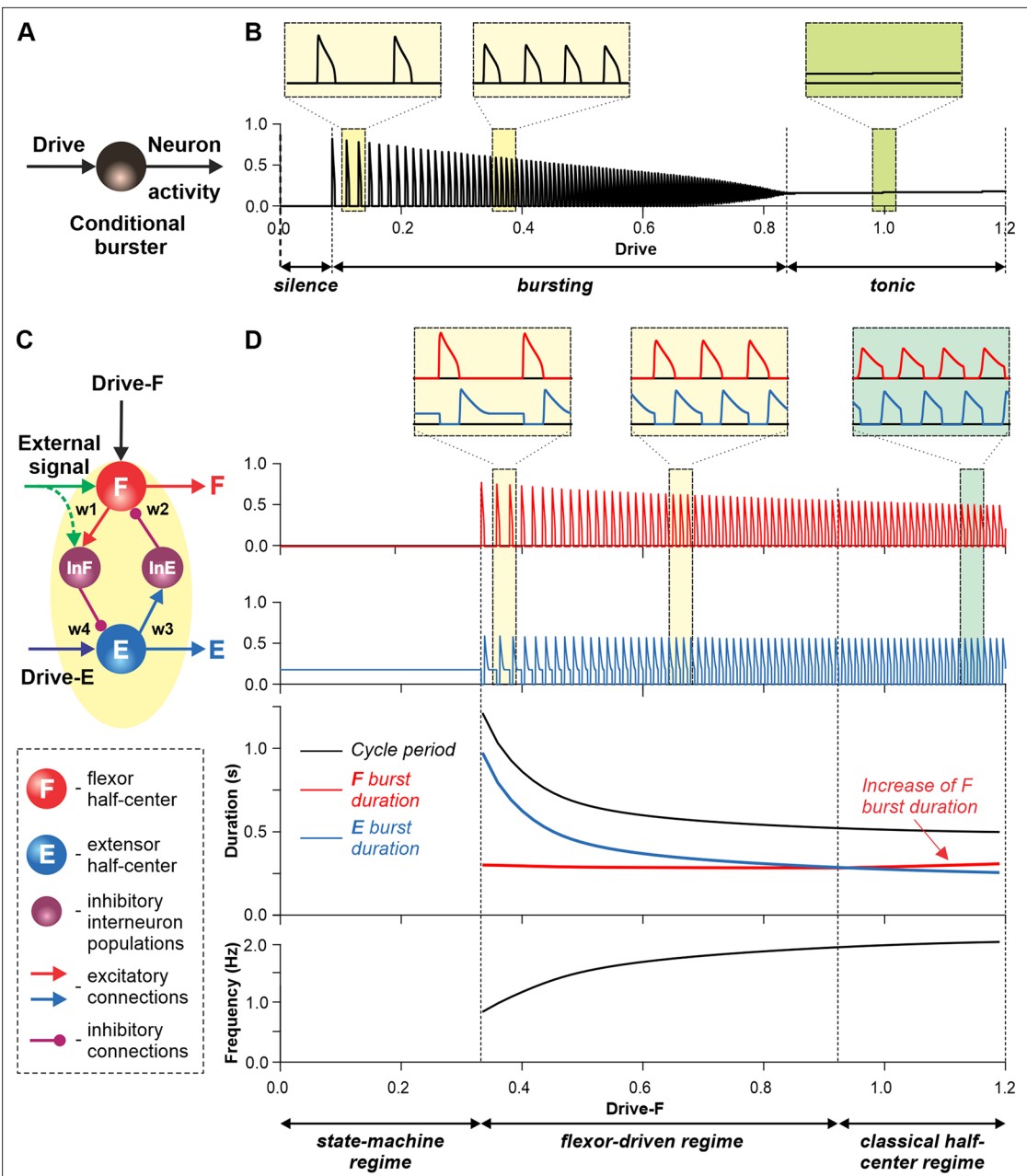

**Figure 3.** Modeling a single conditional burster and a half-center rhythm-generator. (**A**) The behavior of a single $I_{NaP}$-dependent conditional burster. (**B**) Changes in the burster's output when the excitatory input (Drive) progressively increases from 0 to 1.2. With increasing Drive, the initial *silence* state (zero output) at low Drive values changes to an intrinsic *bursting* regime with burst frequency increasing with the Drive value (seen in two left insets), and then to a *tonic* activity (seen in right inset). (**C**) Model of a simple half-center network (rhythm generator, RG) consisting of two conditional bursters/half-centers inhibiting each other through additional inhibitory neurons, InF and InE. The flexor half-center (**F**) receives progressively increasing Drive-F, whereas the extensor half-center (**E**) receives a constant Drive-E keeping it in the regime of tonic activity if uncoupled. (**D**) Model performance. At low Drive-F values, there are no oscillations in the system. This is a *state-machine* regime in which the RG maintains the state of extension, until an external (strong enough) signal arrives to activate the F half-center or to inhibit the E half-center (see green arrows) to release the F half-center from E inhibition allowing it to generate an intrinsic burst. Further increasing the Drive-F releases the F half-center from E inhibition and switches the RG to the bursting regime (see two insets in the middle). In this regime, the E half-center also exhibits bursting activity (alternating with F bursts) due to rhythmic inhibition from the F half-center. This is a *flexor-driven* regime. In this regime, with an increase in Drive-F, the bursting frequency of the RG is increasing (and the oscillation period is decreasing) due to shortening of the extensor bursts with much less reduction in the duration of flexor bursts (see bottom curves and two left insets). Further increasing the excitatory Drive-F leads to a transition of RG operation to a *classical half-center* oscillatory regime, in which none of the half-centers can generate oscillations if uncoupled, and the RG oscillations occur due to mutual inhibition between the half-centers and adaptive properties of their responses. Also in this regime, with an increase of Drive-F, the period of oscillations remains almost unchanged, and the

*Figure 3 continued on next page*

*Figure 3 continued*

duration of flexor bursts increases partly to compensate for the shortening of extensor bursts, which is opposite to the flexor-driven regime (see bottom curves and right inset).

The online version of this article includes the following figure supplement(s) for figure 3:

**Figure supplement 1.** Sensitivity of the single rhythm generator (RG) model to variations of key model parameters.

slow and higher locomotor speeds, respectively (*Rybak et al., 2015*; *Shevtsova et al., 2015*; *Talpalar et al., 2013*). The V3 CINs contribute to left-right synchronization of RG oscillations during a gallop and bound (*Rybak et al., 2015*; *Shevtsova et al., 2015*; *Zhang et al., 2022*; *Danner et al., 2016*; *Danner et al., 2017*; *Danner et al., 2019*).

The proposed spinal network connectome allowed the previous models to reproduce and explain multiple experimental phenomena observed during fictive and real locomotion, including the speed-dependent expression of different gaits and its supraspinal control (*Rybak et al., 2015*; *Shevtsova et al., 2015*; *Zhang et al., 2022*; *Danner et al., 2016*; *Danner et al., 2017*; *Danner et al., 2019*). The molecular/genetic types of neurons in this connectome and their known and suggested interactions were based on studies of intact and mutant mice obtained by optogenetic methods, most of which are not available for studies in cats. We assume that the neural connectome in the spinal cord of mammals is evolutionarily conserved and findings from studies in mice can be used for understanding the neural control of locomotion in other quadrupedal mammals, such as cats.

## Control of spinal locomotor circuits by supraspinal drives and sensory feedback and interactions between them

In our intact model, the frequency of locomotor oscillations (and the speed of locomotion) is primarily controlled by excitatory supraspinal drives to both RGs, particularly to the flexor half-centers (*Figure 4A*). These drives simulate the major descending brainstem pathways to spinal neural circuits. Our previous models also suggested that some supraspinal drives activate/inhibit CINs to modify interlimb coordination and perform gait transitions from left-right alternating (walk, trot) to left-right synchronized (gallop, bound) (*Rybak et al., 2015*; *Shevtsova et al., 2015*; *Zhang et al., 2022*; *Danner et al., 2016*; *Danner et al., 2017*; *Danner et al., 2019*). In the present model, this feature is implemented by supraspinal excitatory signals to V3 CINs (*Figure 4A*).

As described for the single RG model (*Figure 3B*), both left and right RGs start to intrinsically generate rhythmic locomotor activity when the value of supraspinal drives to the flexor half-centers exceeds some threshold. Below this threshold, the RGs can only operate in a state-machine regime that requires an additional excitatory input to the flexor half-centers to initiate flexion. Such additional trigger signals can come from supraspinal structures, such as the motor cortex, or from limb sensory feedback. In the spinal-transected model that lacks supraspinal drives (*Figure 4B*), and in the intact model with low drives necessary for a very slow locomotion, both RGs (specifically, their F half-centers) do not receive sufficient excitation and can only operate in a state-machine regime (see *Figure 3D*), in which the extension to flexion transition requires additional signals that can be provided by limb sensory feedback.

It is well known that the operation of the spinal network during locomotion is also controlled by inputs from primary afferents originating in muscle spindles (groups Ia and II), Golgi tendon organs (group Ib), and different skin mechanoreceptors (reviewed in *Frigon et al., 2021*). Limb sensory feedback controls phase durations and transitions and reinforces extensor activity during stance. We incorporated two types of limb sensory feedback (SF) in our model in a simplified version, both operating during the extensor phase of each limb.

The first feedback, SF-E1, represents an increase of spindle afferent activity from hindlimb flexor muscles as they are stretched with limb extension (an increase of hip angle) during stance (*Klishko et al., 2021*), which increases with increasing treadmill speed. This feedback directly activates the ipsilateral F half-center (*Figure 4*) and promotes the transition from extension to flexion (simulating lift-off and the stance-to-swing transition). The same feedback acting in the model through the V3-E CIN and InE neurons inhibits the contralateral F half-center (*Figure 4*) and promotes the transition from flexion to extension in the contralateral RG. The critical role of hip flexor stretch-related feedback for triggering the stance-to-swing (or extension-to-flexion) transition has been confirmed in many studies

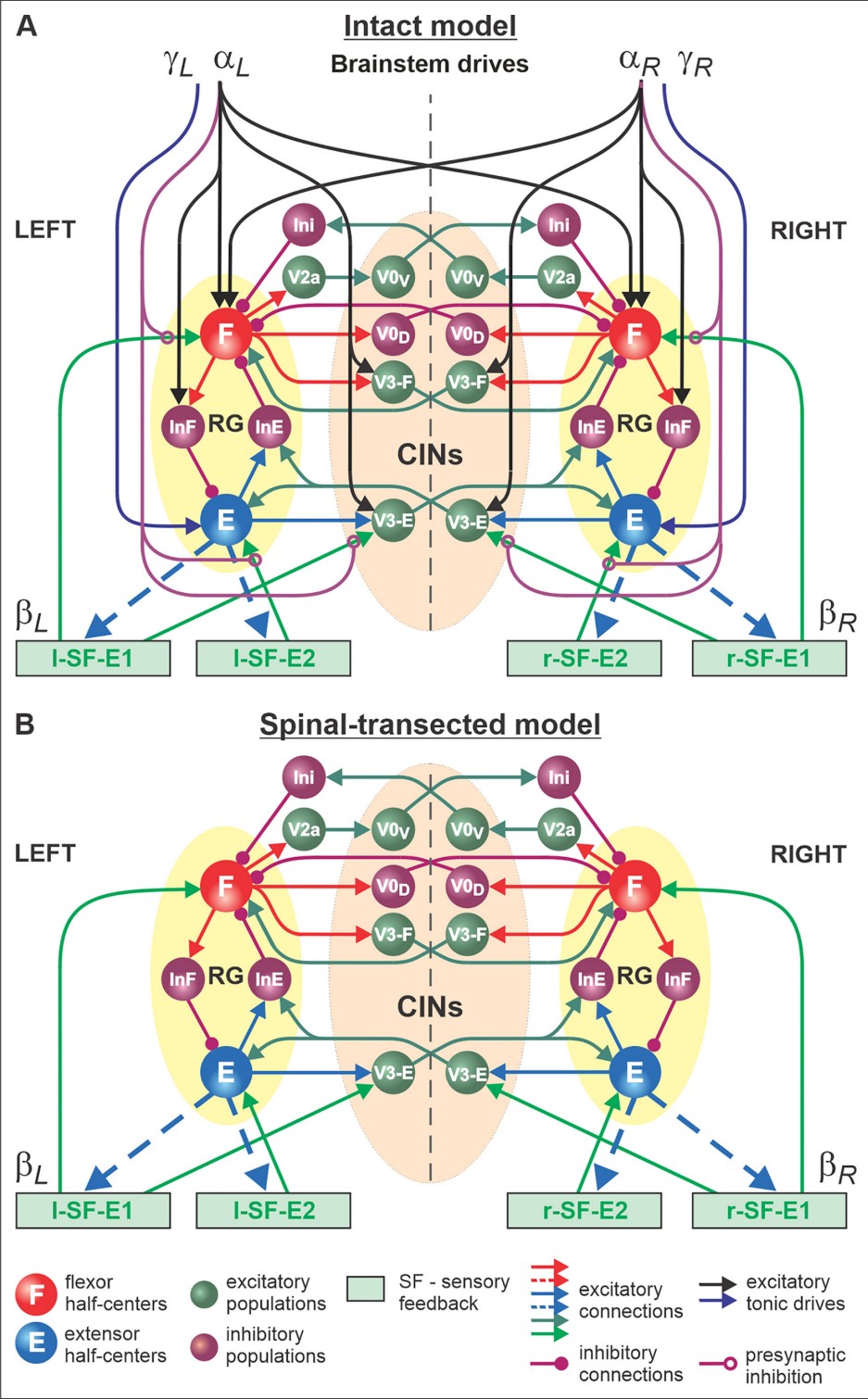

**Figure 4.** Model of spinal circuits controlling treadmill locomotion. (**A**) Model of the intact system ('intact model'). The model includes two bilaterally located (left and right) rhythm generators (RGs) (each is similar to that shown in *Figure 3B*) coupled by (interacting via) several commissural pathways mediated by genetically identified commissural ($V0_D$, $V0_v$, and V3) and ipsilaterally projecting excitatory (V2a) and inhibitory neurons (see text for details). Left and right excitatory supraspinal drives ($\alpha_L$ and $\alpha_R$) provide activation for the flexor half-centers (**F**) of the RGs (ipsi- and contralaterally) and some interneuron populations in the model, as well as for the extensor half-centers (**E**) $\gamma_L$ and $\gamma_R$ ipsilaterally. Two types of feedback (SF-E1 and SF-E2) operating during ipsilateral extension

*Figure 4 continued on next page*

*Figure 4 continued*

affect (excite), respectively, the ipsilateral F and E half-centers, and through V3-E neurons affect contralateral RGs. The SF-E1 feedback depends on the speed of the ipsilateral 'belt' ($\beta_L$ or $\beta_R$) and contributes to extension-to-flexion transition on the ipsilateral side. The SF-E2 feedback activates the ipsilateral E half-center and contributes to 'weight support' on the ipsilateral side. The ipsilateral excitatory drives ($\alpha_L$ and $\alpha_R$) suppress (reduce) the effects of all ipsilateral feedback inputs by presynaptic inhibition. (**B**) Model of the spinal-transected system. All supraspinal drives (and their suppression of sensory feedback) are eliminated from the schematic shown in **A**.

---

in cats during real and fictive locomotion (*Frigon et al., 2021*; *Grillner and Rossignol, 1978*; *Kriellaars et al., 1994*; *Schomburg et al., 1998*; *Pearson, 2004*; *Pearson, 2008*). The role of V3 CINs in transmitting primary afferent activity to the contralateral side was supported by a recent mouse study (*Laflamme et al., 2023*).

The second feedback we incorporated in our model, SF-E2, simulates in a simplified form the involvement of force-dependent Ib positive feedback from limb extensor muscles to the ipsilateral extensor half-center, reinforcing extensor activity and weight support during stance (*Figure 4*). This effect and the role of Ib feedback from extensor afferents has been demonstrated and described in many studies in cats during real and fictive locomotion (*Frigon et al., 2021*; *Duysens and Pearson, 1980*; *Pearson and Collins, 1993*; *Gossard et al., 1994*; *Conway et al., 1987*).

An important feature of our model is how supraspinal drives and limb sensory feedback interact with each other and with spinal circuits. Specifically, limb sensory feedback from each limb receives presynaptic inhibition from the ipsilateral supraspinal drive (*Figure 4A*), which reduces feedback gains in a speed-dependent manner, suppressing the influence of both feedback types. This interaction in the model reflects experimental data on presynaptic and/or direct inhibition of primary afferents by supraspinal signals (*Rudomin and Schmidt, 1999*; *Eccles et al., 1961*; *Fink et al., 2014*; *Lundberg, 1964*). In our model, removing supraspinal drives to simulate the effect of a complete spinal transection also eliminates presynaptic inhibition of all inputs from sensory feedback, increasing the influence of both types of sensory feedback on spinal CPG circuits (*Figure 4B*).

Intact animals walking on a treadmill use visual cues and supraspinal signals to adjust their speed and maintain a fixed position relative to the external space (*Salinas et al., 2017*). In a simplified version (implemented in the model), the movement speed relative to the treadmill belt is mainly controlled by supraspinal drives to the left and right RGs (parameters $\alpha_L$ and $\alpha_R$, see Methods) that define the locomotor oscillation frequency. These drives are automatically adjusted to the speed of the simulated treadmill (parameters $\beta_L$ and $\beta_R$, see Methods). The frequency of oscillations generated by the RGs and the duty factor of the generated pattern are also affected by SF-E1 and SF-E2 acting during the extension phases of each RG, but they are progressively suppressed by supraspinal drives through presynaptic inhibition with increasing speed. Thus, the role of feedback in the control of locomotion decreases with an increase in locomotor frequency defined by the increasing supraspinal drives. In the spinal-transected model, the transition from extension to flexion is fully controlled by limb sensory feedback. The extensor phase duration and the period (and frequency) of oscillations are mainly defined by the rate of SF-E1 increase, which in turn is defined by the speed of the simulated treadmill (parameters $\beta_L$ and $\beta_R$, see Methods).

## Simulation of tied-belt locomotion in intact and spinal cats

We simulated an increase in treadmill speed in the tied-belt condition by the progressive increase of parameters $\beta_L = \beta_R$. As stated above, we assumed that intact cats voluntarily adjust supraspinal drives to the left and right RGs to the treadmill speed to maintain a fixed position relative to the external space. Therefore, in the intact model, the parameters $\alpha_L = \alpha_R$ that characterized supraspinal drives were adjusted to correspond to the parameters $\beta_L = \beta_R$ characterizing treadmill speed. In intact (*Figure 5A*) and spinal-transected (*Figure 5B*) models, changes in cycle and phase durations were qualitatively similar to experimental data (compare with *Figure 1A and B*). Specifically, the oscillation period (cycle duration) decreases with an increasing 'speed' due to a shortening of the extensor phase with a relatively constant flexor phase duration, so that the duty factor approaches or reaches 0.5. Comparison of changes in cycle and phase durations of the intact and spinal-transected models demonstrates similar dynamics of changes (*Figure 5C*), matching corresponding experimental data (*Figure 1C*). It is important to note that in the intact model at 'slow treadmill speeds' (low $\beta_L = \beta_R$) and the corresponding

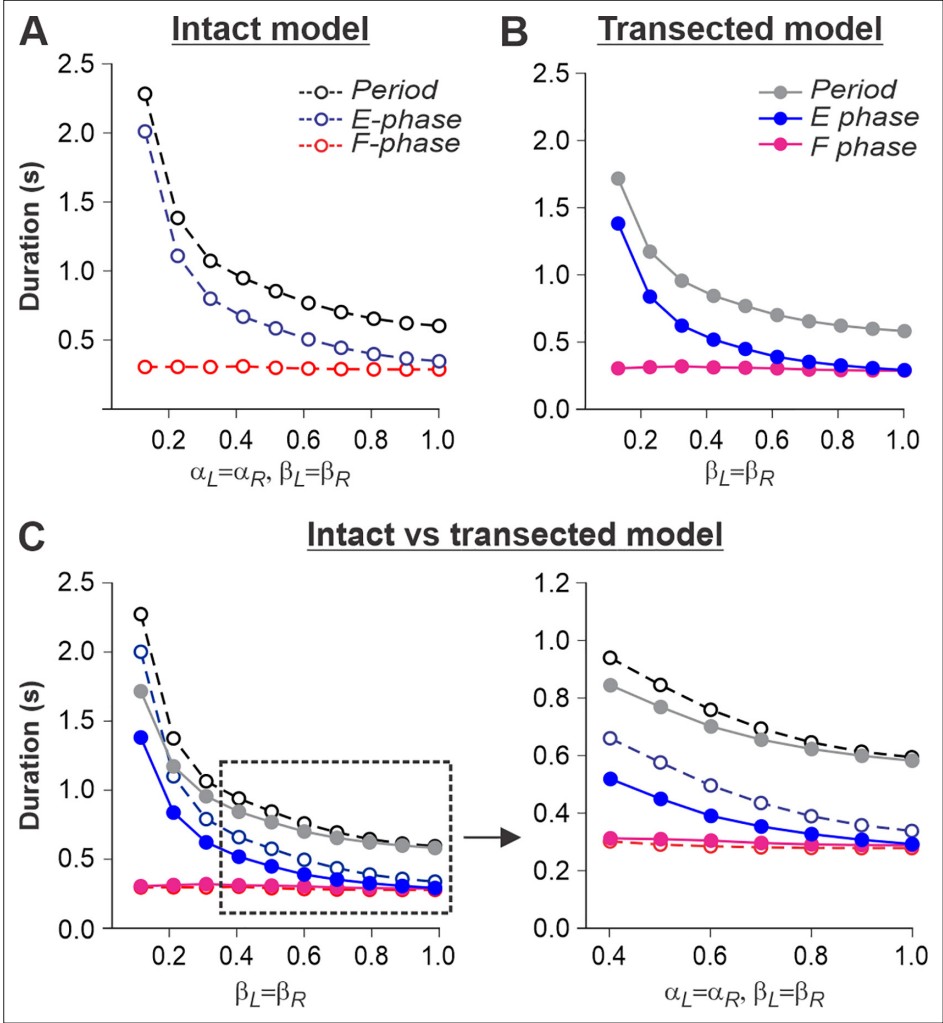

**Figure 5.** Simulation of locomotion on a tied-belt treadmill using intact and transected models. (**A, B**) Changes in the durations of locomotor period and flexor/stance and extensor/swing phases during simulated tied-belt locomotion using the intact (*Figure 4A*) and transected (*Figure 4B*) models with an increasing simulated treadmill speed. (**C**) Superimposed curves from panels (**A**) and (**B**) to highlight differences.

weak supraspinal drives ($\alpha_L=\alpha_R$, approximately at or below 0.4), both RGs are unable to intrinsically generate rhythmic activity and operate in a state-machine regime where switching from the extensor to the flexor phase is mainly controlled by SF-1 and SF-2. At higher values of 'treadmill speeds' and corresponding supraspinal drives, locomotion begins to be mainly controlled by supraspinal drives to the RGs. In this case, the role of limb sensory feedback in the control of phase durations is reduced because of increased presynaptic inhibition by increasing supraspinal drives (see *Figure 5C*). In contrast to the intact model, the transected model has no supraspinal drives to the RGs and can only operate in a state-machine regime at all treadmill speeds. Thus, the transition from extension to flexion and the duration of the extensor phase are entirely controlled by sensory feedback.

## Simulation of split-belt locomotion in intact and spinal cats

For split-belt locomotion, we simulated the speed of the 'slow' RG with a fixed value of $\beta_L = 0.4$ and the 'fast' RG with $\beta_R$ progressively increasing from 0.5 to 1.0. Again, based on our assumption above, the supraspinal drives to the left and right RGs in the intact model were adjusted to the corresponding speeds of the treadmill belts. We set a constant drive $\alpha_L = 0.4$ to the left RG and progressively increased the drive $\alpha_R$ to the right RG from 0.5 to 1.0. We show that in the intact model (*Figure 6A*),

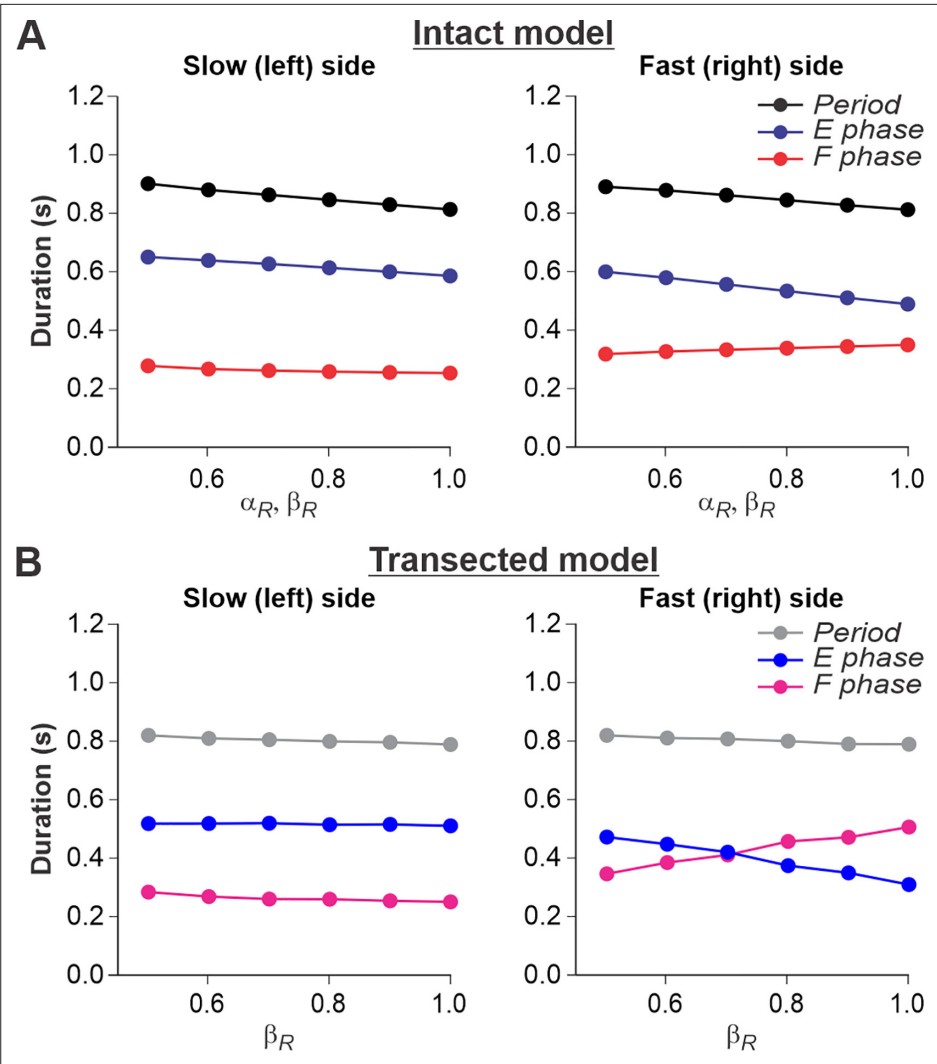

**Figure 6.** Simulation of locomotion on a split-belt treadmill using intact and transected models. (**A**) Changes in the durations of locomotor period and flexor/stance and extensor/swing phases for the left (slow) and right (fast) sides during split-belt treadmill locomotion using the intact model (**Figure 4A**). (**B**) Changes in the same characteristics for the left (slow) and right (fast) sides during the simulation of split-belt treadmill locomotion using the transected model (**Figure 4B**). In both cases, the speed of the simulated left (slow) belt was constant ($\beta_L$ = 0.4) while the speed of the simulated right belt ($\beta_R$) changed from 0.5 to 1.0 with 0.1 increments.

despite differences in the 'speeds' of left and right RGs, the left and right oscillation periods are equal, which corresponds to experimental data (**Figure 2A**).

In the spinal-transected model (**Figure 6B**), changes in cycle and phase durations are similar to those in the intact model and correspond to experimental data (**Figure 2B**). The oscillation periods on the left and right sides are equal, and changes in cycle and phase durations on the left slow side do not change much with a progressive increase of $\beta_R$ from 0.5 to 1.0. The flexor phase duration of the fast RG increases with increasing $\beta_R$, which compensates for the decrease in the extensor phase duration.

An increase of flexion phase duration on the fast side with an increase of speed on that side in the spinal-transected model is much steeper than in the intact model (compare **Figure 6B** with **Figure 6A**) which reproduces corresponding experimental data (see **Figure 2B** vs. **Figure 2A**). The mechanism of this increase in the spinal case (spinal-transected model) is the following. With the increase in speed, the SF-E1 starts accumulating a tonic component that continues acting during flexion. This accumulated tonic activity provides a direct excitation to the ipsilateral F half-center during flexion, which is compensated by inhibition from the contralateral SF-E1 that inhibits the ipsilateral F half-center during

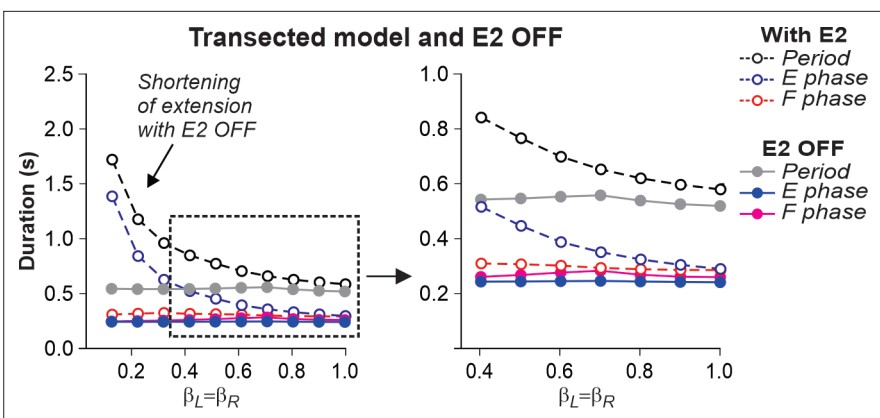

**Figure 7.** Simulation of the effect of removing SF-E2 feedback in the transected model during simulated tied-belt locomotion. Changes in the durations of locomotor period and flexor/stance and extensor/swing phases during simulated tied-belt locomotion using the transected model (*Figure 4B*) after removal of SF-E2 feedback.

the flexor phase (via contralateral V3-E CIN and ipsilateral InE neuron, see *Figure 4*). In tied-belt locomotion (*Figure 5*), this contralateral inhibition acts in balance with the accumulated ipsilateral excitation from SF-E1 and limits excitation of each F half-center during flexion, keeping both RGs within the flexor-driven operating regime with relatively constant flexor phase durations (as in *Figure 3D*, middle part of the graph). In split-belt locomotion, this contralateral inhibition from the slow (left) side remains relatively weak (it is fixed at a value corresponding to the constant slow left-side speed and does not change with the increasing speed on the right fast side), whereas the accumulated excitation on the fast side continues increasing with the speed. Therefore, the F half-center on the fast side becomes overexcited and the RG on the fast side starts operating in the classical half-center regime (like in *Figure 3D*, right part of the graph) with an increasing flexor phase duration as the speed on the fast side increases (*Figure 6B*). A similar mechanism operates in the intact split-belt case (*Figure 6A*), but the resultant increase of the flexor phase duration on the fast side is much weaker due to presynaptic inhibition of left and right SF-E1 feedback.

## Effects of feedback removal on treadmill locomotion

We used our model to simulate the effects of removing limb sensory feedback. The results of these simulations allowed us to formulate modeling predictions that could be tested in future experiments, providing an additional validation to our model. We specifically focused on the simulation of separate removal of SF-E1 feedback, activated with limb extension and involved in the stance-to-swing transition, or SF-E2 feedback involved in the generation of extensor activity and weight support during stance, as well as the removal of both feedback types.

In the transected model, the removal of SF-E1 (with or without SF-E2) prevented locomotion (not shown). Without supraspinal drive and SF-E1, both RGs could not switch to the flexor phase and generate locomotor activity. Following removal of SF-E2 in the transected model, sufficient extensor activity (necessary for 'weight support') could not be developed, leading to an abnormal locomotion with extremely short extensor phase durations (and oscillation periods), with an inability to adjust to treadmill speed (*Figure 7*). In this case, locomotion could be recovered by adding an additional activation of extensor half-centers during stance phases (not shown).

The described above effects of removing limb sensory feedback on locomotion in the spinal-transected model were expected. However, what happens after the removal of limb sensory feedback in the intact model, in which supraspinal drives are present? Removing SF-E1 in the intact model produced interesting results. It considerably increased the duration of the extensor phase (*Figure 8A*). However, the model only demonstrated oscillatory activity starting at some moderate speed, when $\beta_L=\beta_R \geq 0.5$. This required the corresponding supraspinal drives to the RGs to be sufficient ($\alpha_L=\alpha_R \geq 0.5$) to operate in the flexor-driven regime and produce phase transitions without any contributions from SF-E1. In contrast, removing SF-E2 in the intact model reduced the activity of the extensor half-centers, shortening extensor phase durations but preserving oscillations in

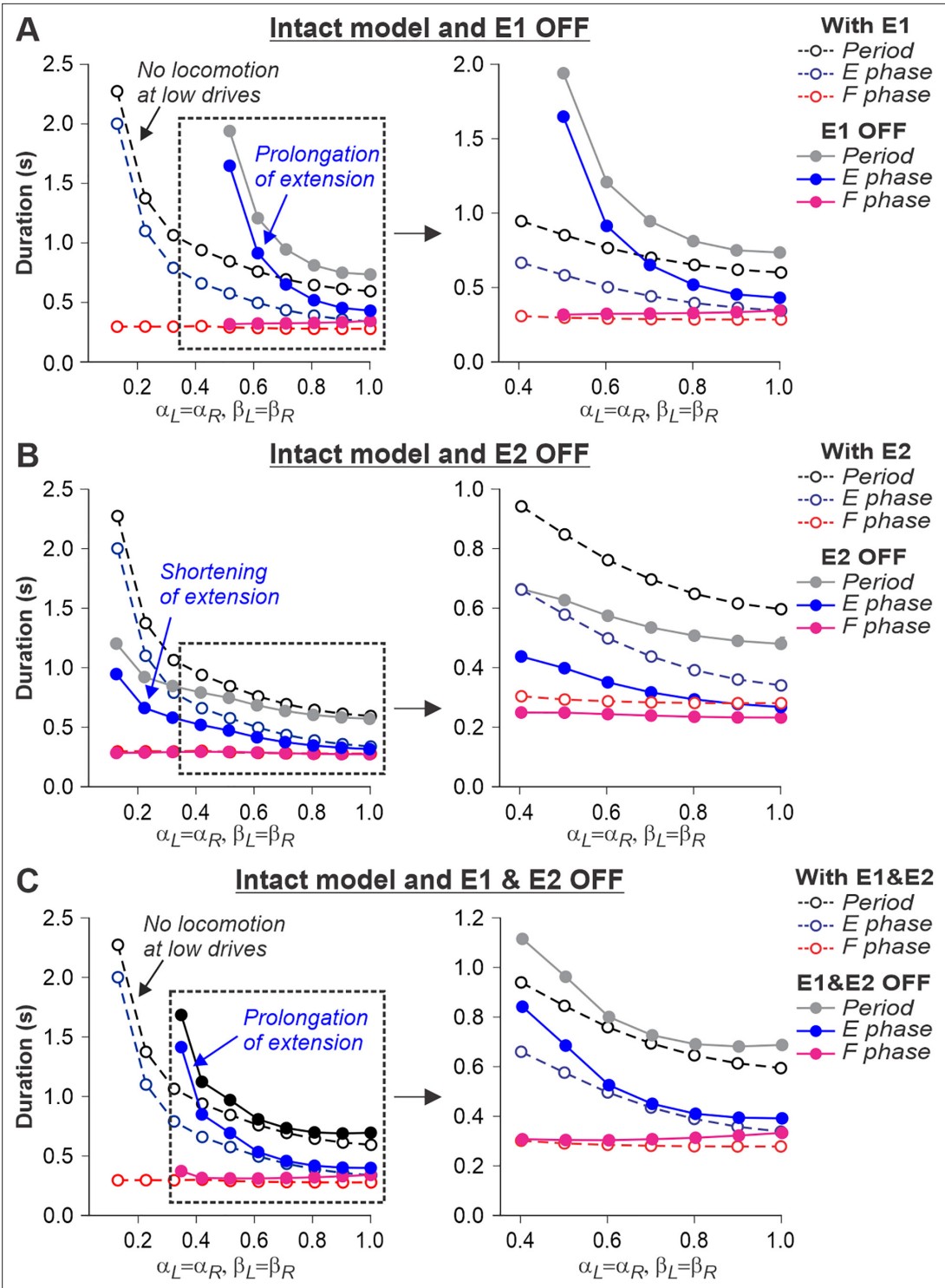

**Figure 8.** Simulation of the effect of removing SF-E1, or SF-E2, or both feedback types in the intact model during simulated tied-belt locomotion. Changes in the durations of locomotor period and flexor/stance and extensor/swing phases during simulated tied-belt locomotion using the intact model (*Figure 4A*) with an increasing simulated treadmill speed after removal of only SF-E1 feedback (**A**), only SF-E2 feedback (**B**), and both feedback types (**C**).

the full range of considered 'speeds' (*Figure 8B*). Therefore, the removal of SF-E2 produced an opposite effect on extensor phase durations compared with removing SF-E1. Finally, removing both feedback types in the intact model shifted the start of oscillatory activity to the left compared to removal of only SF-E1, allowing oscillations to start at $\beta_L=\beta_R \geq 0.35$ (*Figure 8C*). This allows the prediction that below 0.35 m/s, cats with diminished limb sensory feedback can only perform locomotion with patterned step-by-step supraspinal signals to produce phase transitions. This can also explain why intact cats do not usually walk consistently on a treadmill with a speed at or below 0.3 m/s.

## Discussion
### Operation regimes of the spinal locomotor network

Despite decades of research, a commonly accepted definition of the spinal locomotor CPG has not been formulated, and different authors use this term in relation to different entities depending on the context. For instance, the term CPG has been used to designate spinal circuits controlling and coordinating rhythmic movements of all limbs, or controlling only rhythmic movements of a single limb, or even a single joint (*Grillner, 1981*; *Orlovsky et al., 1999*; *McCrea and Rybak, 2008*; *Brown, 1911*; *Brown, 1914*; *Grillner and Zangger, 1975*; *Grillner and Zangger, 1979*; *Rossignol, 1996*; *Grillner, 2006*). Here, we use the term CPG for the spinal circuitry controlling and coordinating all limbs, and the term RG (rhythm generator) for the relatively independent part of the CPG that controls rhythmic movements of a single limb. We consider the CPG as a group of RGs, each controlling a single limb and interacting with each other through commissural and/or propriospinal pathways or circuits (*Frigon, 2017*; *Rybak et al., 2015*; *Shevtsova et al., 2015*; *Zhang et al., 2022*; *Shevtsova et al., 2022*; *Danner et al., 2016*; *Danner et al., 2017*; *Danner et al., 2019*).

Regardless of the exact name (RG or unit burst generator), a common view is that the function of the RG is to generate locomotor-like rhythmic bursting consisting of two major phases, flexion and extension. In each phase, the controlled limb operates in the corresponding functional state, defining the contraction and relaxation of specific sets of muscles. Thus, the RG is not just an oscillator, but rather a system with a dual function: it functions as a *state-machine* (*Dzeladini et al., 2014*; *Spaeth et al., 2020*; *Di Russo et al., 2023*), defining the state and operation of the controlled biomechanical system in each phase independent of the exact transition mechanisms, and as *an oscillator* defining the mechanisms and timing of transitions between states. These transitions may be fully defined by internal properties of the RG and/or require a contribution from external inputs, such as supraspinal signals or somatosensory feedback, that control or adjust the timing of transitions.

Here, we described a relatively simple model of a half-center RG. We identified three operation regimes: *state-machine*, *flexor-driven*, and *classical half-center*. In an intact system, at slow speeds (≤0.35 m/s), the spinal network operates in a state regime and requires external inputs for phase transitions, which can come from limb sensory feedback and/or volitional inputs (e.g. from the motor cortex). At higher speeds and greater supraspinal drives, the spinal network switches to a flexor-driven regime before transitioning to a classical half-center regime at higher speeds/drives. Following spinal transection, the spinal network can only operate in the state regime and entirely depends on limb sensory feedback for phase transitions.

Our modeling results also resolve a potential contradiction between the classical half-center and flexor-driven concepts of spinal RG operation. According to the *classical half-center concept*, both flexor and extensor half-centers are necessary for generating rhythmic activity (*McCrea and Rybak, 2008*; *Brown, 1911*; *Brown, 1914*). An alternative, *flexor-driven concept* was proposed by Pearson and Duysens ('swing generator model'), where the flexor half-center is intrinsically rhythmic, unlike the extensor half-center, which exhibits rhythmic activity due to rhythmic inhibition from the flexor half-center (*Pearson and Duysens, 1976*; *Duysens et al., 2013*). Here, we follow our previous modeling studies (*Latash et al., 2020*; *Ausborn et al., 2018*) and show that these concepts do not contradict each other, but rather relate to different regimes of RG operation. With an increase of excitatory drive to an RG, the rhythmogenic mechanism changes from a flexor-driven to a classical half-center mechanism (*Figure 3*). This transition explains the increase of the flexor phase duration of the fast hindlimb or RG during split-belt treadmill locomotion in intact and spinal cats (*Figures 2 and 6*).

## Intrinsic spinal rhythm-generating versus reflex-based mechanisms of locomotion

Following Maurice Philippson's view formulated from studies in spinal dogs (*Philippson, 1905*), Charles Sherrington proposed that locomotion in decerebrate and spinal cats was generated by chains of reflexes (*Sherrington, 1910a*; *Sherrington, 1910b*). Yet, Thomas Graham Brown clearly showed that an intrinsic spinal mechanism alone, without somatosensory feedback, could generate rhythmic alternating activity (*Brown, 1911*), which later became the CPG concept. Our results support both concepts, depending on locomotor speed and the state of the animal. Based on our simulations, we suggest that in spinal cats at any speed and in intact cats at low speeds, the spinal network operates in a state-machine regime and requires some sensory feedback to locomote, consistent with Philippson's/Sherrington's viewpoint. In contrast, at higher (moderate) speeds (≥0.4 m/s), when a supraspinal drive is sufficient, the spinal network can intrinsically generate the basic locomotor activity controlling locomotion, which supports Brown's concept.

Our results also suggest an important difference in the control of slow exploratory and faster escape locomotion (*Caggiano et al., 2018*; *Ferreira-Pinto et al., 2018*; *Kim et al., 2017*; *Branco and Redgrave, 2020*). Based on our predictions, slow (conditionally exploratory) locomotion is not 'automatic' but requires volitional (e.g. cortical) signals to trigger step-by-step phase transitions because the spinal network operates in a state-machine regime. In contrast, locomotion at moderate to high speeds (conditionally escape locomotion) occurs automatically under the control of spinal rhythm-generating circuits receiving supraspinal drives that define locomotor speed, unless voluntary modifications or precise stepping are required to navigate complex terrain (*Drew et al., 1996*; *Beloozerova et al., 2010*).

We also used our model to simulate and predict the effects of removing limb sensory feedback on locomotion. As expected, the spinal-transected model failed to generate locomotion without SF-E1 or normal locomotion without SF-E2 because the spinal network operates in a state-machine regime (*Figure 7*). The essential role of somatosensory feedback after spinal cord injury is well-known (*Frigon et al., 2021*; *Takeoka, 2020*; *Takeoka and Arber, 2019*; *Takeoka et al., 2014*; *Bouyer and Rossignol, 2003*; *Rossignol and Frigon, 2011*). However, in the intact model, limb sensory feedback, particularly from hip flexor afferents (SF-E1), is also required for locomotion and phase transitions at slow speeds to compensate for low supraspinal drives (*Figure 8*).

Another important implication of our results relates to the recovery of walking in movement disorders, where the recovered pattern is generally very slow. For example, in people with spinal cord injury, the recovered walking pattern is generally less than 0.1 m/s and completely lacks automaticity (*Wagner et al., 2018*; *Angeli et al., 2018*; *Gill et al., 2018*). Based on our predictions, because the spinal locomotor network operates in a state-machine regime at these slow speeds, subjects need volition, additional external drive (e.g. epidural spinal cord stimulation) or to make use of limb sensory feedback by changing their posture to perform phase transitions. Another paper commenting on observations in Parkinsonian patients also proposed that different CPG control systems operate at slow and fast speeds (*Duysens and Nonnekes, 2021*).

The concept of a spinal locomotor CPG in humans is strongly supported by a variety of experimental and clinical evidence (*Duysens and Van de Crommert, 1998*; *Danner et al., 2015*; *Minassian et al., 2017*; *Yang et al., 2004*). In human walking, the basic output of the CPG, reflected in the EMG pattern of leg muscles, is maintained even at very slow speeds, albeit with differences in amplitudes and some bursting patterns from slow to fast walking speeds (*den Otter et al., 2004*; *Hof et al., 2002*). This is consistent with the idea that the same CPG provides the basic motor program from slow to fast speeds but that additional inputs or drives are required. The idea of a flexible arrangement of CPG circuits has also been proposed to explain the generation of locomotion and other rhythmic motor behaviors (e.g. scratching, paw shaking) in various species, such as cats, turtles, zebrafish, and rats (*Frigon and Gossard, 2010*; *Berkowitz et al., 2010*; *Berkowitz and Hao, 2011*; *Juvin et al., 2007*; *Harris-Warrick, 2011*; *Parker et al., 2018*).

## Model limitations and future directions

In this study, we developed a simplified model of the neural control of cat hindlimb locomotion in tied- and split-belt conditions to simulate and compare the locomotion of intact and spinal cats. The major limitation of the present model is the lack of biomechanical elements simulating multi-joint

limbs and muscles. Our study and analysis were specifically focused on the proposed organization and operation of spinal CPG circuits, including left-right interactions within these circuits, and their control by supraspinal drives and limb sensory feedback. Note that the term 'supraspinal drive in our model is used to represent supraspinal inputs providing both electrical and neuromodulator effects on spinal neurons to increase their excitability, which disappears after spinal transection. The other limitation of the present model is that it does not consider the possibility that afferent feedback can provide some constant level of excitation to the RG circuits after spinal transection, which can partly compensate for the lack of supraspinal drive and hence affect (shift) the timing of transitions between the considered regimes. We will consider this issue in the future. Another important limitation is that we did not consider possible mechanisms of plasticity following spinal transection. We know that the spinal connectome and sensorimotor interactions change after spinal cord injury (*Rossignol and Frigon, 2011*; *Frigon and Rossignol, 2006*). The precise nature of these changes is not well understood. Nevertheless, we believe that changes (increase) in the gain of sensory feedback, which in the model results from eliminating presynaptic inhibition, in a real situation may include a considerable contribution from plastic changes during the recovery period. However, even with the account of possible plastic changes during the recovery period, the functional role of the increased gain of sensory feedback in spinal-transected animals remains the same independent of the exact mechanisms for this increase (release from presynaptic inhibition, plastic changes during recovery, currently unknown mechanisms, or any combination of the above).

Although the model was based on simplifications and some assumptions, it reproduced and provided explanations for experimental results, such as the main locomotor characteristics (cycle and phase durations) at different speeds and left-right speed differences during tied-belt and split-belt locomotion. We are currently applying our model to simulate cat locomotion following incomplete spinal cord injury (lateral hemisection) during tied-belt and split-belt locomotion based on our recent experimental data (*Audet et al., 2023*; *Mari et al., 2024*; *Lecomte et al., 2022*). Using our model, this will allow us to remove supraspinal drives on the hemisected side and determine how the spinal network controls locomotion. We also plan to incorporate a full biomechanical model of the limbs to investigate the neuromechanical control of quadrupedal locomotion and its recovery following incomplete spinal cord injury (*Markin, 2016*; *Prilutsky, 2016*).

## Methods
### Experimental data
For this modeling study, we used previously published data obtained from intact and spinal cats during tied-belt (equal left-right speeds) and split-belt (different left-right speeds) locomotion (*Frigon et al., 2015*; *Frigon et al., 2017*; *Latash et al., 2020*). No new animals were used here. In those studies, all procedures were approved by the Animal Care Committee of the Université de Sherbrooke (Protocol 442–18) in accordance with policies and directives of the Canadian Council on Animal Care. Intact cats performed quadrupedal locomotion whereas spinal cats performed hindlimb-only locomotion with the forelimbs on a stationary platform. Intact and spinal cats performed tied-belt locomotion from 0.4 to 1.0 m/s and from 0.1 to 1.0 m/s, respectively, with 0.1 m/s increments. As stated earlier, intact cats cannot perform consistent quadrupedal tied-belt locomotion at or below 0.3 m/s. For split-belt locomotion, the slow belt (left) was 0.4 m/s while the fast belt (right) stepped at speeds of 0.5–1.0 m/s in 0.1 m/s increments.

### Modeling formalism and model parameters
In our model, we considered spinal circuits as a network of interacting neural populations. Each population is described by an activity-based neuron model (and is sometimes called a *neuron* in the text), in which the dependent variable $V$ represents an average population voltage and the output $f(V)$ $(0 \leq f(V) \leq 1)$ represents the average or integrated population activity at the corresponding average voltage (*Latash et al., 2020*; *Danner et al., 2016*; *Danner et al., 2017*; *Ausborn et al., 2019*; *Rubin et al., 2009*; *Ermentrout, 1994*). This description allows an explicit representation of ionic currents, specifically of the persistent (slowly inactivating) sodium current (*Rubin et al., 2009*), which was proposed to be responsible for generating intrinsic bursting activity in the spinal cord (*Brocard et al., 2013*; *Rybak et al., 2006a*; *Rybak et al., 2006b*; *Tazerart et al., 2007*; *Tazerart et al., 2008*; *Brocard*

et al., 2010; **McCrea and Rybak, 2007**; **Zhong et al., 2012**). Assuming that neurons within each population switch between silence and active spiking in a generally synchronized way, the dynamics of the average voltages are represented within a conductance-based framework used for a single neuron description, but without fast membrane currents responsible for spiking activity.

The dynamics of $V$ for flexor and extensor half-centers, considered as $I_{NaP}$-dependent conditional bursters, are described by the following differential equation:

$$C \cdot \frac{dV}{dt} = -I_{NaP} - I_L - I_{SynE} - I_{SynI}. \tag{1}$$

For other (non-bursting) populations, $V$ is described as:

$$C \cdot \frac{dV}{dt} = -I_L - I_{SynE} - I_{SynI}. \tag{2}$$

The output function $f(V)$ transvers $V$ to the integrated population output and is defined as follows:

$$f(V) = \begin{cases} 0, \text{ if } V < V_{\text{thr}}; \\ (V - V_{\text{thr}})/(V_{\text{max}} - V_{\text{thr}}), \text{ if } V_{\text{thr}} \leq V < V_{\text{max}}; \\ 1, \text{ if } V \geq V_{\text{max}}. \end{cases} \tag{3}$$

In **Equations 1 and 2**, $C$ is the membrane capacitance, $I_{NaP}$ is the persistent sodium current, $I_L$ is the leakage current, $I_{SynE}$ and $I_{SynI}$ represent excitatory and inhibitory synaptic currents, respectively. The leakage current was described as:

$$I_L = g_L \cdot (V - E_L) \tag{4}$$

where $g_L$ is the leakage conductance and $E_L$ represents the leakage reversal potential. The persistent sodium current in the flexor and extensor half-centers is described as:

$$I_{NaP} = \bar{g}_{NaP} \cdot m(V) \cdot h \cdot (V - E_{Na}), \tag{5}$$

where $\bar{g}_{NaP}$ is the maximal conductance and $E_{Na}$ is the sodium reversal potential. Its voltage-dependent activation, $m(V)$, is instantaneous, and its steady state is described as follows:

$$m(V) = m_\infty(V) = \left\{ 1 + \exp\left[ \left( V - V_{1/2,m} \right) / k_m \right] \right\}^{-1}. \tag{6}$$

The slow $I_{NaP}$ inactivation was modeled by the following differential equation:

$$\tau_h(V) \cdot \frac{dh}{dt} = h_\infty(V) - h, \tag{7}$$

where $h_\infty(V)$ represents inactivation steady state and $\tau_h(V)$ is the inactivation time constant with maximal value $\tau_{max}$.

$$h_\infty(V) = \left\{ 1 + \exp\left[ \left( V - V_{1/2,h} \right) / k_h \right] \right\}^{-1}; \tag{8}$$

$$\tau_h(V) = \tau_{max}/\cosh\left[ \left( V - V_{1/2,\tau} \right) / k_\tau \right]. \tag{9}$$

In **Equations 6, 8, and 9**, $V_{1/2}$ and $k$ represent the half-voltage and slope of the corresponding variables ($m$, $h$, and $\tau$). Excitatory and inhibitory synaptic currents ($I_{SynE}$ and $I_{SynI}$) for population $i$ were described by:

$$I_{SynE,i} = g_{SynE} \cdot \left\{ \sum_j \left[ S\left( w_{ji} \right) \cdot f(V_j) \right] + D_i + SF_i^{E1} + SF_i^{E2} \right\} \cdot \left( V_i - E_{SynE} \right); \tag{10}$$

$$I_{SynI,i} = g_{SynI} \cdot \left\{ \sum_j \left[ S\left( w_{ji} \right) \cdot f\left( V_j \right) \right] \right\} \cdot \left( V_i - E_{SynE} \right), \tag{11}$$

**Table 1.** Connection weights.

**Weights of connections between neurons, $w_{ji}$**

| Source neuron | Target neuron |
| --- | --- |
| F | InF (3.6), V0$_D$ (2.5), V2a (2.5), V3-F (1.0) |
| E | InE (3.6), V3-E (0.3) |
| InF | E (–8.0) |
| InE | F (–4.0) |
| V2a | V0$_V$ (2.5) |
| V0$_V$ | c-Ini (2.5) |
| Ini | F (–2.0) |
| V0$_D$ | c-F (–6) |
| V3-E | c-E (0.1), c-InE (1.5) |
| V3-F | c-F (0.1) |

**Weights of connections from drives to neurons: $k_i^{\alpha ipsi}, k_i^{\alpha contra}, k_i^{\gamma}$**

| Source drive | Target neuron |
| --- | --- |
| $\alpha_{ipsi}$[∈ 0; 1.0] | F (0.45), InF(0.5), V3-E(0.1), V3-F(0.1) |
| $\alpha_{contra}$[∈ 0; 1.0] | F (0.15) |
| $\gamma$=1.0 | E (2.0) |

**Weights of connections from feedback sources to neurons: $k_i^{E1}, k_i^{E2}$**

| Synaptic input from feedback | Target neuron |
| --- | --- |
| E1 | F (1.4), V3-E (1.2) |
| E2 | E (2.0) |

**Presynaptic inhibition of feedback connections by $\alpha_{ipsi}$: $k_{PSi}^{E1}, k_{PSi}^{E2}$**

| Synaptic input from feedback | Target feedback connection |
| --- | --- |
| E1 | F (4.0), V3-E (0.5) |
| E2 | E (2.0) |

c-contralateral.

where $g_{SynE}$ and $g_{SynI}$ are synaptic conductances and $E_{SynE}$ and $E_{SynI}$ are the reversal potentials of the excitatory and inhibitory synapses, respectively; $w_{ji}$ is the synaptic weight from population $j$ to population $t$ ($w_{ji} > 0$ for excitatory connections and $w_{ji} < 0$ for inhibitory connections).

$$S(x) = \begin{cases} x, \text{if } x \geq 0, \\ 0, \text{if } x < 0. \end{cases} \qquad (12)$$

The weights of connections $w_{ji}$ between populations for the main model (**Figure 4**) are shown in **Table 1**. $D_i$ in **Equation 10** for the main model (**Figure 4**), represents the total excitatory drive to population $i$:

$$D_i = k_i^{\alpha ipsi} \cdot \alpha_{ipsi} + k_i^{\alpha contra} + k_i^{\gamma} \cdot \gamma, \qquad (13)$$

where $\alpha_{ipsi}$ and $\alpha_{contra}$ are the ipsi- and contralateral excitatory drives, respectively, which depending on the side, represent left $\alpha_L$ and right $\alpha_R$ drives, respectively; parameter $\gamma$ represents the constant

drive to the ipsilateral (left or right) extensor half-center; $k_i^{\alpha ipsi}$, $k_i^{\alpha contra}$ and $k_i^{\gamma}$ define the weights of these drives to population $i$.

$SF_i^{E1}$ and $SF_i^{E2}$ in **Equation 10** for the main model (**Figure 4**), define the effect of ipsilateral sensory feedback E1 (or SF-E1, see **Figure 4**) and E2 (or SF-E2), respectively, to population $i$. The gain of each feedback to population $i$ ($k_i^{E1}$ and $k_i^{E2}$) is suppressed (reduced) by the ipsilateral drive $\alpha_{ipsi}$ ($\alpha_L$ or $\alpha_R$ depending on the side):

$$SF_i^{E1} = k_i^{E1} \cdot E1 / \left(1 + k_{PSi}^{E1} \cdot \alpha_{ipsi}\right);$$
(14)

$$SF_i^{E2} = k_i^{E2} \cdot E2 / \left(1 + k_{PSi}^{E2} \cdot \alpha_{ipsi}\right),$$
(15)

where $k_{PSi}^{E1}$ and $k_{PSi}^{E2}$ are the weights of presynaptic inhibition by $\alpha_{ipsi}$ of feedback inputs E1 and E2, respectively, to population $i$ (see **Figure 4A**).

E1 feedback (SF-E1) represents an increase in the activity of length-dependent hip flexor afferents during limb extension. In our model, it is described as:

$$\tau_{E1} \cdot \frac{dE1}{dt} = \left[k_{E1} \cdot \beta_{ipsi}^{1.25} \cdot (t - t_0)\right] \cdot sign\left[f\left(V_E\right) - thr\right] - E1,$$
(16)

$$\text{where } \tau_{E1} = \begin{cases} 0, \text{ if} \dfrac{dE1}{dt} \geq 0, \\ \tau_1, \text{ if} \dfrac{dE1}{dt} < 0, \end{cases}$$

$$sign[y] = \begin{cases} 1, \text{if } y > 0, \\ 0, \text{if } y \leq 0, \end{cases}$$

where $k_{E1}$ is the parameter of E1 feedback; $\beta_{ipsi}$ is the parameter characterizing the speed of the ipsilateral treadmill belt (left $\beta_L$ or right $\beta_R$), $f(V_E)$ is the output of the ipsilateral E half-center, $thr$ is the threshold, $t_0$ is the starting time of the ipsilateral extensor burst and $t$ is time (both in ms).

E2 feedback (SF-E2) that represents excitatory feedback from load-dependent afferents of limb extensor muscles to the extensor half-center during extension, described as follows:

$$\tau_{E2} \cdot \frac{dE2}{dt} = \left[E2_o - k_{E2} \cdot \beta_{ipsi}^{1.25} \cdot (t - t_0)\right] \cdot sign\left[f\left(V_E\right) - thr\right] - E2,$$
(17)

$$\text{where } \tau_{E2} = \begin{cases} \tau_2, \text{ if } \dfrac{dE2}{dt} \geq 0, \\ 0, \text{if} \dfrac{dE2}{dt} < 0, \end{cases}$$

where $E2o$ and $k_{E2}$ are the parameters of SF-E2.

Variables $\alpha_L$ and $\alpha_R$ characterize left and right supraspinal drives to RGs and define the frequency of locomotor oscillations. The variables $\beta_L$ and $\beta_R$ define the speeds of the left and right treadmill belts. Intermediate coefficients were used to make correspondence between $\alpha$ and $\beta$ values and between their values and the speed of real treadmill belts in the experiments. For modeling of intact locomotion, we changed left and right $\alpha$ values to simulate 'voluntary' locomotion in the intact model via a corresponding adjustment of supraspinal drives. For modeling of locomotion in the spinal-transected model, supraspinal drives were set to 0 and we only manipulated left and right $\beta$ values simulating speeds of treadmill belts.

The following values of model parameters were used for the main model (**Figure 4**): $C = 20\ pF$; $g_L = 4\ nS$ for RG half-centers and $g_L = 1\ nS$ for all other neurons; $\bar{g}_{NaP} = 4.4\ nS$; $g_{synE} = g_{synI} = 1\ nS$; $E_{Na} = 50.0\ mV$; $E_{EsynE} = -10\ mV$; $E_{EsynI} = -75\ mV$; $V_{thr} = -50\ mV$; $V_{max} = 0\ mV$; $V_{1/2,m} = -40.0\ mV; k_m = -6\ mV; V_{1/2,h} = -45.0\ mV; k_h = 4\ mV; \tau_{max} = 500\ ms; V_{1/2,\tau} = -45\ mV; k_\tau = 20\ mV$; $E2o=2.5$; $k_{E1} = 3.12 \cdot 10^{-3} ms^{-1}$; $k_{E2} = 0.5 \cdot 10^{-3} ms^{-1}$; $\tau_1 = 500\ ms$; $\tau_2 = 300\ ms$; $thr = 0.1$. All connection weights for the main model (**Figure 4**) are specified in **Table 1**.

In the models shown in **Figure 3**: $E_L = -64.0\ mV$; $w_1 = w_3 = 3.6$; $w_2 = 3.0; w_4 = 5.0$; Drive-E=1.2. All other parameters are the same as in the main model, listed above.

## Simulations, data analysis, and availability

All simulations were performed using the custom neural simulation package NSM 2.5.7. The simulation package was previously used for models of spinal circuits (*McCrea and Rybak, 2008*; *Shevtsova et al., 2015*; *Zhang et al., 2022*; *Shevtsova et al., 2022*; *Danner et al., 2019*; *Rybak et al., 2006a*; *Rybak et al., 2006b*; *McCrea and Rybak, 2007*; *Zhong et al., 2012*; *Shevtsova, 2016*). Differential equations were solved using the exponential Euler integration method with a step size of 0.1 ms. Simulation results were saved as ASCII files and represented the output functions for half-centers recorded with a precision of 0.1 ms.

The simulation results were processed using custom Matlab scripts (The Mathworks, Inc, Matlab 2023b). To assess model behavior, the activities of flexor and extensor half-centers were used to determine the onsets and offsets of flexor and extensor bursts and to calculate flexor and extensor phase durations and oscillation periods. The timing of onsets and offsets of flexor and extensor bursts was determined at a threshold level of 0.05. The oscillation period was defined as the duration between two consecutive burst onsets in the left extensor half-center. The flexor and extensor phase durations and oscillation periods were averaged over the duration of the simulation for each value of the parameter $\alpha$. Duration of individual simulations depended on the value of parameter $\alpha$ to robustly estimate average values of burst durations and oscillation periods. For each $\alpha$ value, we omitted the first two transitional cycles to allow stabilization of model variables. Flexor and extensor phase durations and oscillation periods were plotted against the parameters $\alpha$ and $\beta$.

The simulation package NSM 2.5.7, the model configuration file necessary to create and run simulations, and the custom Matlab scripts are available at https://github.com/RybakLab/nsm (copy archived at *RybakLab, 2024*).

# Additional information

### Funding

| Funder | Grant reference number | Author |
|---|---|---|
| National Institutes of Health | R01 NS110550 | Ilya A Rybak<br>Natalia A Shevtsova<br>Sergey N Markin<br>Boris I Prilutsky<br>Alain Frigon |
| National Science Foundation | 2024414 | Boris I Prilutsky |

The funders had no role in study design, data collection and interpretation, or the decision to submit the work for publication.

### Author contributions

Ilya A Rybak, Conceptualization, Formal analysis, Supervision, Funding acquisition, Validation, Visualization, Methodology, Writing - original draft, Project administration, Writing – review and editing; Natalia A Shevtsova, Data curation, Software, Formal analysis, Validation, Investigation, Visualization, Methodology, Writing – review and editing; Sergey N Markin, Resources, Software, Formal analysis, Writing – review and editing; Boris I Prilutsky, Formal analysis, Investigation, Writing – review and editing; Alain Frigon, Data curation, Validation, Investigation, Visualization, Methodology, Writing - original draft, Writing – review and editing

### Author ORCIDs

Ilya A Rybak ⓘ https://orcid.org/0000-0003-3461-349X
Natalia A Shevtsova ⓘ https://orcid.org/0000-0002-1971-9707
Boris I Prilutsky ⓘ https://orcid.org/0000-0003-0499-3890
Alain Frigon ⓘ https://orcid.org/0000-0002-9259-2706

## Ethics

All experimental procedures were approved by the Animal Care Committee of the Université de Sherbrooke (Protocol 442-18) in accordance with policies and directives of the Canadian Council on Animal Care.

Reviewer #1 (Public review): https://doi.org/10.7554/eLife.98841.3.sa1

Reviewer #2 (Public review): https://doi.org/10.7554/eLife.98841.3.sa2

Reviewer #3 (Public review): https://doi.org/10.7554/eLife.98841.3.sa3

Author response https://doi.org/10.7554/eLife.98841.3.sa4

# Additional files

## Supplementary files

• MDAR checklist

## Data availability

The current manuscript is a computational study. The simulation package used, the model configuration file necessary to create and run simulations, and the custom Matlab scripts are available at https://github.com/RybakLab/nsm, copy archived at *RybakLab, 2024*.

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
