## [Editor Report · eLife assessment]

This **fundamental** state-of-the-art modeling study explores neural mechanisms underlying walking control in cats, demonstrating the probability of three different states of operation of the spinal circuitry generating locomotion at different speeds. The authors' biophysical modeling sufficiently reproduces and provides explanations for experimental data on how the locomotor cycle and phase durations depend on treadmill walking speed and points to new principles of circuit functional architecture and operating regimes underlying how spinal circuits interact with supraspinal signals and limb sensory feedback signals to produce different locomotor behaviors at different speeds, which are major unresolved problems in the field. The modeling evidence is **compelling**, especially in advancing our understanding of locomotion control mechanisms and will interest neuroscientists studying the neural control of movement.

---

## [Referee Report · Reviewer #1 (Public review)]

Summary:

It is suggested that for each limb, the RG (rhythm generator) can operate in three different regimes: a non-oscillating state-machine regime and a flexor driven and a classical half-center oscillatory regime. This means that the field can move away from the old concept that there is only room for the classic half-center organization

Strengths:

A major benefit of the present paper is that a bridge was made between various CPG concepts ("a potential contradiction between the classical half-center and flexor-driven concepts of spinal RG operation"). Another important step forward is the proposal about the neural control of slow gait ("at slow speeds ({less than or equal to} 0.35 m/s), the spinal network operates in a state regime and requires external inputs for phase transitions, which can come from limb sensory feedback and/or volitional inputs (e.g. from the motor cortex")).

Weaknesses:

Some references are missing

---

## [Referee Report · Reviewer #2 (Public review)]

Summary:

The biologically realistic model of the locomotor circuits developed by this group continues to define the state of the art for understanding spinal genesis of locomotion. Here the authors have achieved a new level of analysis of this model to generate surprising and potentially transformative new insights. They show that these circuits can operate in three very distinct states and that, in the intact spinal cord, these states come into successive operation as the speed of locomotion increases. Equally important, they show that in spinal injury, the model is "stuck" in the low-speed "state machine" behavior.

Strengths:

There are many strengths for the simulations results presented here. The model itself has been closely tuned to match a huge range of experimental data and this has a high degree of plausibility. The novel insight presented here, with the three different states, constitutes a truly major advance in the understanding of neural genesis of locomotion in spinal circuits. The authors systematically consider how the states of the model relate to presently available data from animal studies. Equally important, they provide a number of intriguing and testable predictions. It is likely that these insights are the most important achieved in the past 10 years. It is highly likely proposed multi-state behavior will have a transformative effect on this field.

Weaknesses:

I have no major weaknesses. A moderate concern is that the authors should consider some basic sensitivity analyses to determine if the 3-state behavior is especially sensitive to any of the major circuit parameters-e.g., connection strengths in the oscillators.

---

## [Referee Report · Reviewer #3 (Public review)]

General Comments

This work probes the control of walking in cats at different speeds and different states (split-belt and regular treadmill walking). Since the time of Sherrington there has been ongoing debate on this issue. The authors provide modeling data showing that they could reproduce data from cats walking on a specialized treadmill allowing for regular and split-belt walking. The data suggest that a non-oscillating state-machine regime best explains slow walking - where phase transitions are handled by external inputs into the spinal network. They then show at higher speeds a flexor-driven and then a classical half-center regime dominates. In spinal animals, it appears that a non-oscillating state-machine regime best explains the experimental data. The model is adapted from their previous work and raises interesting questions regarding the operation of spinal networks, that, at low speeds, challenge assumptions regarding central pattern generator function. This is an outstanding study which will be of general interest to the neuroscience community.

Strengths

The study has several strengths. Firstly the detailed model has been well established by the authors and provides details that relate to experimental data such as commissural interneurons (V0c and V0d), along with V3 and V2a interneuron data. Sensory input along with descending drive is also modelled and moreover the model reproduces many experimental data findings. Moreover, the idea that sensory feedback is more crucial at lower speeds, also is confirmed by presynaptic inhibition increasing with descending drive. The inclusion of experimental data from split-belt treadmills, and the ability of the model to reproduce findings here is a definite plus.

Weaknesses

Conceptually, this is a compelling study which provides interesting modeling data regarding the idea that the network can operate in different regimes, especially at lower speeds. The modelling data speaks for itself, but on the other hand, sensory feedback also provides generalized excitation of neurons which in turn project to the CPG. That is they are not considered part of the CPG proper. The authors have discussed this possibility in their revised paper.

---

## [Author Response]

The following is the authors’ response to the original reviews.

**Public Reviews:**

**Reviewer #1 (Public Review):**
Summary:It is suggested that for each limb the RG (rhythm generator) can operate in three different regimes: a non-oscillating state-machine regime, and in a flexor driven and a classical half-center oscillatory regime. This means that the field can move away from the old concept that there is only room for the classic half-center organizationStrengths:A major benefit of the present paper is that a bridge was made between various CPG concepts ("a potential contradiction between the classical half-center and flexor-driven concepts of spinal RG operation"). Another important step forward is the proposal about the neural control of slow gait ("at slow speeds ({less than or equal to} 0.35 m/s), the spinal network operates in a state regime and requires external inputs for phase transitions, which can come from limb sensory feedback and/or volitional inputs (e.g. from the motor cortex")).Weaknesses:Some references are missing

We thank the Reviewer for the thoughtful and constructive comments. We have added additional text to meet the specific Reviewer’s recommendations and several references suggested by the Reviewer.

**Reviewer #2 (Public Review):**
Summary:The biologically realistic model of the locomotor circuits developed by this group continues to define the state of the art for understanding spinal genesis of locomotion. Here the authors have achieved a new level of analysis of this model to generate surprising and potentially transformative new insights. They show that these circuits can operate in three very distinct states and that, in the intact cord, these states come into successive operation as the speed of locomotion increases. Equally important, they show that in spinal injury the model is "stuck" in the low speed "state machine" behavior.Strengths:There are many strengths for the simulation results presented here. The model itself has been closely tuned to match a huge range of experimental data and this has a high degree of plausibility. The novel insight presented here, with the three different states, constitutes a truly major advance in the understanding of neural genesis of locomotion in spinal circuits. The authors systematically consider how the states of the model relate to presently available data from animal studies. Equally important, they provide a number of intriguing and testable predictions. It is likely that these insights are the most important achieved in the past 10 years. It is highly likely proposed multi-state behavior will have a transformative effect on this field.Weaknesses:I have no major weaknesses. A moderate concern is that the authors should consider some basic sensitivity analyses to determine if the 3 state behavior is especially sensitive to any of the major circuit parameters - e.g. connection strengths in the oscillators or?

We thank the Reviewer for the thoughtful and constructive comments. The sensitivity analysis has been included as Supplemental file.

**Reviewer #3 (Public Review):**
Summary:This work probes the control of walking in cats at different speeds and different states (split-belt and regular treadmill walking). Since the time of Sherrington there has been ongoing debate on this issue. The authors provide modeling data showing that they could reproduce data from cats walking on a specialized treadmill allowing for regular and split-belt walking. The data suggest that a non-oscillating state-machine regime best explains slow walking - where phase transitions are handled by external inputs into the spinal network. They then show at higher speeds a flexor-driven and then a classical halfcenter regime dominates. In spinal animals, it appears that a non-oscillating state-machine regime best explains the experimental data. The model is adapted from their previous work, and raises interesting questions regarding the operation of spinal networks, that, at low speeds, challenge assumptions regarding central pattern generator function. This is an interesting study. I have a few issues with the general validity of the treadmill data at low speeds, which I suspect can be clarified by the authors.Strengths:The study has several strengths. Firstly the detailed model has been well established by the authors and provides details that relate to experimental data such as commissural interneurons (V0c and V0d), along with V3 and V2a interneuron data. Sensory input along with descending drive is also modelled and moreover the model reproduces many experimental data findings. Moreover, the idea that sensory feedback is more crucial at lower speeds, also is confirmed by presynaptic inhibition increasing with descending drive. The inclusion of experimental data from split-belt treadmills, and the ability of the model to reproduce findings here is a definite plus.Weaknesses:Conceptually, this is a very useful study which provides interesting modeling data regarding the idea that the network can operate in different regimes, especially at lower speeds. The modelling data speaks for itself, but on the other hand, sensory feedback also provides generalized excitation of neurons which in turn project to the CPG. That is they are not considered part of the CPG proper. In these scenarios, it is possible that an appropriate excitatory drive could be provided to the network itself to move it beyond the state-machine state - into an oscillatory state. Did the authors consider that possibility? This is important since work using L-DOPA, for example, in cats or pharmacological activation of isolated spinal cord circuits, shows the CPG capable of producing locomotion without sensory or descending input.We thank the Reviewer for the thoughtful and constructive comments. We have added additional texts, references, and discussed the issues raised by the Reviewer. Particularly, in section “Model limitations and future directions” we now admit that afferent feedback can provide some constant level excitation to the RG circuits after spinal transection which can partly compensate for the lack of supraspinal drive and hence affect (shift) the timing of transitions between the considered regimes. We mentioned that this is one of the limitations of the present model. The potential effects of neuroactive drugs, like DOPA, on CPG circuits after spinal transection were left out because they are outside the scope of the present modeling studies.
**Recommendations for the authors:**

**Reviewer #1 (Recommendations For The Authors):**
specific feedback to the authors:Nevertheless, there are some minor points, worth considering.Link to HUMAN DATAHere the authors may be interested to know that human data supports their proposal. This is relevant since there is ample evidence for the operation of spinal CPG's in humans (Duysens and van de Crommert,1998). The present model predicts that the basic output of the CPG remains even at very slow speeds, thus leading to similarity in EMG output. This prediction fits the experimental data (den Otter AR, Geurts AC, Mulder T, Duysens J. Speed related changes in muscle activity from normal to very slow walking speeds. Gait Posture. 2004 Jun;19(3):270-8). To investigate whether the basic CPG output remains basically the same even at very slow speeds (as also predicted by the current model), humans walked slowly on a treadmill (speeds as slow as 0.28 m s−1). Results showed that the phasing of muscle activity remained relatively stable over walking speeds despite substantial changes in its amplitude. Some minor additions were seen, consistent with the increased demands of postural stability. Similar results were obtained in another study: Hof AL, Elzinga H, Grimmius W, Halbertsma JP. Speed dependence of averaged EMG profiles in walking. Gait Posture. 2002 Aug;16(1):78-86. doi:10.1016/s0966-6362(01)00206-5. PMID: 12127190.These authors wrote: "The finding that the EMG profiles of many muscles at a wide range of speeds can be represented by addition of few basic patterns is consistent with the notion of a central pattern generator (CPG) for human walking". The basic idea is that the same CPG can provide the motor program at slow and fast speeds but that the drive to the CPG differs. This difference is accentuated under some conditions in pathology, such as in Parkinson's Kinesia Paradoxa. It was argued that the paradox is not really a paradox but is explained as the CPGs are driven by different systems at slow and at fast speeds (Duysens J, Nonnekes J. Parkinson's Kinesia Paradoxa Is Not a Paradox. Mov Disord. 2021 May;36(5):1115-1118. doi: 10.1002/mds.28550. Epub 2021 Mar 3. PMID: 33656203.)These ideas are well in line with the current proposal ("Based on our predictions, slow (conditionally exploratory) locomotion is not "automatic", but requires volitional (e.g. cortical) signals to trigger stepby-step phase transitions because the spinal network operates in a state-machine regime. In contrast, locomotion at moderate to high speeds (conditionally escape locomotion) occurs automatically under the control of spinal rhythm-generating circuits receiving supraspinal drives that define locomotor speed, unless voluntary modifications or precise stepping are required to navigate complex terrain").As mentioned in the present paper, other examples exist from pathology ("...Another important implication of our results relates to the recovery of walking in movement disorders, where the recovered pattern is generally very slow. For example, in people with spinal cord injury, the recovered walking pattern is generally less than 0.1 m/s and completely lacks automaticity 77-79. Based on our predictions, because the spinal locomotor network operates in a state-machine regime at these slow speeds, subjects need volition, additional external drive (e.g., epidural spinal cord stimulation) or to make use of limb sensory feedback by changing their posture to perform phase transitions"). As mentioned above, another example is provided by Parkinson's disease. The authors may also be interested in work on flexible generators in SCI: Danner SM, Hofstoetter US, Freundl B, Binder H, Mayr W, Rattay F, Minassian K. Human spinal locomotor control is based on flexibly organized burst generators. Brain. 2015 Mar;138(Pt 3):577-88. doi: 10.1093/brain/awu372. Epub 2015 Jan 12. PMID: 25582580; PMCID: PMC4408427.

We thank the reviewer for these additional and interesting insights. We added a new paragraph in the Discussion to bolster the link with human data that includes references suggested by the Reviewer.

CHAIN OF REFLEXESIt reads: "... in opposition to the previously prevailing viewpoint of Charles Sherrington 21,22 that locomotion is generated through a chain of reflexes, i.e., critically depends on limb sensory feedback (reviewed in 23)." This is correct but incomplete. The reference cited (23: Stuart, D.G. and Hultborn, H, "Thomas Graham Brown (1882--1965), Anders Lundberg (1920-), and the neural control of stepping," Brain Res. Rev. 59(1), 74-95 (2008)) actually reads: "Despite the above findings, the doctrinaire position in the early 1900s was that the rhythm and pattern of hind limb stepping movements was attributable to sequential hind limb reflexes. According to Graham Brown (1911c) this viewpoint was largely due to the arguments of Sherrington and a Belgian physiologist, Maurice Philippson (1877-1938). Philippson studied stepping movements in chronically maintained spinal dogs, using techniques he had acquired in the Strasbourg laboratory of the distinguished German physiologist, Friedrich Goltz (1834-1902). He also analyzed kinematically moving pictures of dog locomotion, which had been sent to him by the renowned French physiologist, Etienne-Jules Marey (1830-1904). Philippson (1905) certainly presented arguments explaining his perception of how sequential spinal reflexes contributed to the four phases of the step cycle (see Fig. 1 in Clarac, 2008). In retrospect, it is likely that Graham Brown was correct in attributing to Philippson and Sherrington the then-prevailing viewpoint that reflexes controlled spinal stepping. It is puzzling, nonetheless, that far less was said then and even now about Philippson's belief that the spinal control was due to a combination of central and reflex mechanisms (Clarac, 2008),4,5 4 We are indebted to François Clarac for drawing to our attention Philippson's statement on p. 37 of his 1905 article that "Nos expériences prouvent d'une part que la moelle lombaire séparée du reste de l'axe cérébro-spinal est capable de produire les mouvements coordonnés dans les deux types de locomotion, trot et gallop. [Our experiments prove that one side of the spinal cord separated from the cerebro-spinal axis is able to produce coordinated movements in two types of locomotion, trot and gallop]." Then, on p. 39 Philippson (1905) states that "Nous voyons donc, en résumé que la coordination locomotrice est une fonction exclusivement médullaire, soutenue d'une part par des enchainements de réflexes directs et croisés, dont l'excitant est tantot le contact avec le sol, tantot le mouvement même du membre. [In summary, we see that locomotor coordination is an exclusive function of the spinal cord supported by a sequencing of direct and crossed reflexes, which are activated sometimes by contact with the ground and sometimes even by leg movement]. A coté de cette coordination basée sur des excitations périphériques, il y a une coordination centrale provenant des voies d'association intra-médullaires. [In conjunction with this peripherally excited coordination, there is a central coordination arising from intraspinal pathways]." (The English translations have also been kindly supplied by François Clarac.) Clearly, Philippson believed in both a central spinal and a reflex control of stepping! 5 In part 1 of his 1913/1916 review Graham Brown discussed Philippson's 1905 article in much detail (pp. 345-350 in Graham Brown, 1913b). He concludes with the statement that "... Philippson die wesentlichen Factoren des Fortbewegungsaktes in das exterozeptive Nervensystem verlegt. Er nimmt an, dass die zyklischen Bewegungen automatisch durch äussere Reize erhalten werden, welche in sich selbst thythmisch als Folge der Reflexakte welche sie selbst erzeugen, wiederholt werden. [Philippson assigns the important factors of the act of locomotion to the exteroceptive nervous system. He assumes that the cyclic movements are automatically maintained by external stimuli which, by themselves, are rhythmically repeated as a consequence of the reflexive actions that they generate themselves]." (English translation kindly supplied by Wulfila Gronenberg). This interpretation clearly ignores Philippson's emphasis on a central spinal component in the control of stepping..... "Hence it is a simplification to give all credits to Sherrington and ignoring the role of Philippson concerning the chain of reflexes idea.

We again thank the Reviewer for these additional and interesting insights. We added the Philippson (1905) and Clarac (2008) references. The important contribution of Philippson is now indicated.

GTO Ib feedbackIt reads: "This effect and the role of Ib feedback from extensor afferents has been demonstrated and described in many studies in cats during real and fictive locomotion 2,57-59."These citations are appropriate but it is surprising to see that the Hultborn contribution is limited to the Gossard reference while the even more important earlier reference to Conway et al is missing (Conway BA, Hultborn H, Kiehn O. Proprioceptive input resets central locomotor rhythm in the spinal cat. Exp Brain Res. 1987;68(3):643-56. doi: 10.1007/BF00249807. PMID: 3691733).

Yes, the Conway et al. reference has been added.

Other speciesThe authors may also look at other species. The flexible arrangement of the CPGs, as described in this article, is fully in line with work on other species, showing cpg networks capable to support gait, but also scratching, swimming ..etc (Berkowitz A, Hao ZZ. Partly shared spinal cord networks for locomotion and scratching. Integr Comp Biol. 2011 Dec;51(6):890-902. doi: 10.1093/icb/icr041. Epub 2011 Jun 22. PMID: 21700568. Berkowitz A, Roberts A, Soffe SR. Roles for multifunctional and specialized spinal interneurons during motor pattern generation in tadpoles, zebrafish larvae, and turtles. Front Behav Neurosci. 2010 Jun 28;4:36. doi: 10.3389/fnbeh.2010.00036. PMID: 20631847; PMCID: PMC2903196.)Similar ideas about flexible coupling can also be found in: Juvin L, Simmers J, Morin D. Locomotor rhythmogenesis in the isolated rat spinal cord: a phase-coupled set of symmetrical flexion extension oscillators. J Physiol. 2007 Aug 15;583(Pt 1):115-28. doi: 10.1113/jphysiol.2007.133413. Epub 2007 Jun 14. PMID: 17569737; PMCID: PMC2277226. Or zebrafish: Harris-Warrick RM. Neuromodulation and flexibility in Central Pattern Generator networks. Curr Opin Neurobiol. 2011 Oct;21(5):685-92. doi: 10.1016/j.conb.2011.05.011. Epub 2011 Jun 7. PMID: 21646013; PMCID: PMC3171584.

We added a sentence in the Discussion along with supporting references.

StandingIn the view of the present reviewer, the model could even be extended to standing in humans. It reads: "at slow speeds ({less than or equal to} 0.35 m/s), the spinal network operates in a state regime and requires external inputs"; similarly (personal experience) when going from sit to stand: as soon as weight is over support, extension is initiated and the body raises, as one would expect when the extensor center is activated by reinforcing load feedback, replacing GTO inhibition (Faist M, Hoefer C, Hodapp M, Dietz V, Berger W, Duysens J. In humans Ib facilitation depends on locomotion while suppression of Ib inhibition requires loading. Brain Res. 2006 Mar 3;1076(1):87-92. doi:)

Yes, we agree that the model could be extended to standing and the transition from standing to walking is particularly interesting. However, for this paper, we will keep the focus on locomotion over a range of speeds.

**Reviewer #2 (Recommendations For The Authors):**
The presentation is exceedingly well done and very clear.A moderate concern is that the authors do not make use of the capacity of computer simulations for sensitivity analyses. Perhaps these have been previously published? In any case, the question here is whether the 3 state behavior is especially sensitive to excitability of one of the main classes of neurons or a crucial set of connections.

The sensitivity analysis has been made and included as Supplemental file.

Minor point. I have but two minor points. A bit more explanation should be provided for the use of the terms "state machine" to describe the lowest speed state. Perhaps this is a term from control theory? In any case, it is not clear why this is term is appropriate for a state in which the oscillator circuits are "stuck" in a constant output form and need to be "pushed" by sensory input.

Yes, we now provide a definition in the Introduction.

Minor point: it is of course likely that neuromodulation of multiple types of spinal neurons occurs via inputs that activate G protein coupled receptors. These types of inputs are absent from the model, which is fine, but some sort of brief discussion should be included. One possibility is to note that the circuit achieves transitions between different states without the need for neuromodulatory inputs. This appears to me to be a very interesting and surprising insight.

In section “Model limitations and future directions” in the Discussion**,** we now mention that the term “supraspinal drive” in our model is used to represent supraspinal inputs providing both electrical and neuromodulator effects on spinal neurons increasing their excitability, which disappear after spinal transection.” We think that it is so far too early to simulate the exact effects of the descending neuromodulation, since there is almost no data on the effect of different modulators on specific types of spinal interneurons.

**Reviewer #3 (Recommendations For The Authors):**
Minor CommentsPage numbers would be useful.AbstractFollowing spinal transection, the network can only operate in a state-machine regime. This is a bit strong since it applies to computational data. Clarify this statement.

We agree. Sentence has been changed to: “Following spinal transection, the model predicts that the spinal network can only operate in the state-machine regime.”

IntroductionIntro - "This is somewhat surprising...". It gives the impression that spinal cats are autonomously stable on the belt. They are stabilized by the experimenter.

The text has been changed to: “This is somewhat surprising because intact and spinal cats rely on different control mechanisms. Intact cats walking freely on a treadmill engage vision for orientation in space and their supraspinal structures process visual information and send inputs to the spinal cord to control locomotion on a treadmill that maintains a fixed position of the animal relative to the external space. Spinal cats, whose position on the treadmill relative to the external space is fixed by an experimenter, can only use sensory feedback from the hindlimbs to adjust locomotion to the treadmill speed.”

"Cannot consistently perform treadmill locomotion" - likely a context-dependent result. Certainly, cats can do this easily off a treadmill - stalking, for example. Perhaps somewhere, mention that treadmill locomotion is not entirely similar to overground locomotion.

We completely agree. Stalking is an excellent example showing that during overground locomotion slow movements (and related phase transitions) can be controlled by additional voluntary commands from supraspinal structures, which differs from simple treadmill locomotion, performing out of specific goalor task-dependent contexts. Based on this, we suggest a difference between a relatively slow (exploratory-type, including stalking) and relatively fast (escape-type) overground locomotion. We added the following sentence to the introduction:” This is evidently context dependent and specific for the treadmill locomotion as cats, humans and other animals can voluntarily decide to perform consistent overground locomotion at slow speeds.”

The authors introduce the concept of the state machine regime. In my opinion, this could use some more explanation and citations to the literature. Was it a term coined by the authors, or is there literature reinforcing this point?

This is a computer science and automata theory term that has already been used in descriptions of locomotion (see our references in the 2nd paragraph of Discussion). We added a definition and corresponding references in the Introduction.

In terms of sensory feedback, particularly group II input, it would be interesting to calculate if the conduction delay to the spinal cord at higher speeds would have a certain cutoff point at which it would no longer be timed effectively for phase transitions. This could reinforce your point.

This is an interesting proposition but it is unlikely to be a factor over the range of speeds that we investigated (0.1 to 1.0 m/s). Assuming that group II afferents transmit their signals to spinal circuits at a latency of 10-20 ms, this is more than enough time to affect phase transitions, even at the highest speed considered. This might be a factor at very high speeds (e.g. galloping) or in small animals with high stepping frequencies.

Results.The assertion that intact cats are inconsistent in terms of walking at slow speeds needs to be bolstered. For example, if a raised platform were built for a tray of food, would the intact cat consistently walk at slower speeds and eat? I suspect so. By the same token, would they walk slowly during bipedal walking? It is pretty easy to check this. Also, reports from the literature show differential effects of runway versus treadmill gait analysis, specifically when afferent input is removed.

The Reviewer is correct that raising a platform for a food tray or even having intact cats walk with their hindlimbs only (with forelimbs on a stationary platform) may allow for consistent stepping at slow speeds (0.1 – 0.3 m/s). However, this effectively removes voluntary control of locomotion and makes the pattern more automatic (spinal + limb sensory feedback). These examples provide additional specific contexts, and we have already mentioned (see above) that slow locomotion of intact cat is context dependent.

"We believe that intact animals walking on a treadmill..." Citations for this? Certainly, this is not a new point.

No, this is not new. We changed the sentence and added a reference to the statement: “Intact animals walking on a treadmill use visual cues and supraspinal signals to adjust their speed and maintain a fixed position relative to the external space with reference to Salinas et al. (Salinas, M.M., Wilken, J M, and Dingwell, J B, "How humans use visual optic flow to regulate stepping during walking," Gait. Posture. 57, 15-20, 2017).

The presentation of the results is somewhat disjointed. The intact data is presented for tied and splitbelt results, but this is not addressed explicitly until figure 4. Would it not be better to create a figure incorporating both intact and modelling data and present the intact data where appropriate?

We tried to do this initially, but this way required changing the style of the whole paper and we decided against this idea. Therefore, we prefer to keep the presentation of results as it is now.

Regarding the role of sensory feedback being especially important at low speeds, it is interesting that egr3+ mice (lacking spindle input) show an inability to walk at high speeds >40 cm/s but can walk at lower speeds (up to 7 cm/s) (Takeoka et al 2014). Similar findings were found with a lesion affecting Group I afferents in general (Takeoka and Arber 2019). Also, Grillner and colleagues show that cats can produce fictive locomotion in the absence of sensory input.

In the Takeoka experiments it is difficult to assess the effect of removing somatosensory feedback because animals can simply decide to not step at higher speeds to avoid injury. Their mice deprived of somatosensory feedback can walk at slow speeds, likely thanks to voluntary commands, and cannot do so at higher speeds because (1) maybe somatosensory feedback is indeed necessary and/or (2) because they feel threatened because of impaired posture and poor control in general. In other words, they choose to not walk at faster speeds to avoid injury.

Fictive locomotion by definition is without phasic somatosensory feedback as the animals are curarized or studies are performed in isolated spinal cord preparations. Depending on the preparation, pharmacology or brainstem stimulation is required to evoke fictive locomotion. If animals are deafferented, pharmacology or brainstem stimulation are required to induce fictive locomotion to offset the loss of spinal neuronal excitability provided by primary afferents. At the same time, our preliminary analysis of old fictive locomotion data in the University of Manitoba Spinal Cord center (Drs. Markin and Rybak had an official access to these data base during our collaboration with Dr. David McCrea) has shown that the frequency of stable fictive locomotion in cats usually exceeded 0.6 - 0.7 Hz, which approximately corresponds to the speed above 0.3 - 0.4 m/s. These data and estimation are just approximate; they have not been statistically analyzed and published and hence have not been included in our paper.

Discussion. The statement that sensory feedback is required for animals to locomote may need to be qualified. Animals need some sensory feedback to locomote is perhaps better. For example, lesion studies by Rossignol in the early 2000s showed that cutaneous feedback from the paw was seemingly quite critical (in spinal cats). Also, see previous comments above.

We changed this to: “… requires some sensory feedback to locomote, …”

FiguresFigure 1C. This figure is somewhat confusing. If intact cats do not walk (arrow), how are the data for swing and stance computed? Also raw traces would be useful to indicate that there is variability. Also, while duration is useful, would you not want to illustrate the co-efficient of variation as well as another way to show that the stepping pattern was inconsistent?

This is probably a misunderstanding. The left panel of Fig. 1C superimposes data of intact cats from panel A (with speed range from 0.4 m/s to 1.0 m/s) and data from spinal cats from panel B (with speed range from 0.1 m/s and 1.0 m/s). Therefore, the left part of this left panel 1C with speed range from 0.1 m/s to 0.4 m/s (pointed out by the arrow) corresponds only to spinal cats (not to intact cats). The standard deviations of all measurements are shown. All these figures were reproduced from the previous publications. We did not apply new statistical analysis to these previously published data/figures.

Figure 4. 'All supraspinal drives (and their suppression of sensory feedback) are eliminated from the schematic shown in A. ' However, it is labelled 'brainstem drives,' which is confusing. Moreover, many of the abbreviations are confusing. Do you need l-SF-E1 in the figure, or could you call it 'Feedback 1' and then refer to l-SF-E1 in the legend? The same goes for βr, etc. Can they move to the legend?

In the intact model (Fig. 4A), we have supraspinal drives (𝛼𝐿 and 𝛼𝑅, and 𝛾𝐿 and 𝛾𝑅), some of which provide presynaptic inhibition of sensory feedback (SF-E1 and SF-E2) as shown in Fig. 4A. In spinaltransected model (Fig. 4B), the above brainstem drives and their effects (presynaptic inhibition) on both feedback types are eliminated (therefore, there is no label “Brainstem drives in Fig. 4B). Also, we do not see a strong reason to change the feedback names, since they are explained in the text.

I appreciate the detail of these figures, but they are difficult to conceptualize. They are useful in the context of 3C. Perhaps move this figure to supplementary and then show the proposed schematics for the system operating at slow, medium, and fast speeds in a replacement figure?

We apologize for the resistance, but we would like to keep the current presentation.

There is a lack of raw data (models or experimental) data reinforcing the figures. I would add these to all figures, which would nicely complement the graphs.

These raw data can be found in the cited manuscripts. It would be the same figures.